# Uncertainty-Aware Gaussian Map for Vision-Language Navigation

**Jianzhe Gao**[1]     **Rui Liu**[1]     **Yuxuan Xu**[1]     **Tongtong Cao**[2]     **Yingxue Zhang**[3]
**Zhanguang Zhang**[3]     **Sida Peng**[4]     **Yi Yang**[1]     **Wenguan Wang**[1*]

[1]The State Key Lab of Brain-Machine Intelligence, Zhejiang University
[2]Department of Foundation model, 2012 Labs, Huawei     [3]Noah's Ark Lab, 2012 Labs, Huawei
[4]School of Software Technology, Zhejiang University
https://github.com/Gaozzzz/Uncertainty-Aware-VLN

## Abstract

Vision-Language Navigation (VLN) requires an agent to navigate 3D environments following natural language instructions. During navigation, existing agents commonly encounter perceptual uncertainty, such as insufficient evidence for reliable grounding or ambiguity in interpreting spatial cues, yet they typically ignore such information when predicting actions. In this work, we explicitly model three forms of perceptual uncertainty (*i.e.*, geometric, semantic, and appearance uncertainty) and integrate them into the agent's observation space to enable informed decision-making. Concretely, our agent first constructs a Semantic Gaussian Map (SGM), composed of differentiable 3D Gaussian primitives initialized from panoramic observations, that encodes both the geometric structure and semantic content of the environment. On top of SGM, geometric uncertainty is estimated through variational perturbations of Gaussian position and scale to assess structural reliability; semantic uncertainty is captured by perturbing Gaussian semantic attributes to reveal ambiguous interpretations; and appearance uncertainty is characterized by Fisher Information, which measures the sensitivity of rendered observations to Gaussian-level variations. These uncertainties are incorporated into SGM, extending it into a unified 3D Value Map, which grounds them as affordances and constraints that support reliable navigation. Comprehensive evaluations across multiple VLN benchmarks show the effectiveness of our agent.

## 1 Introduction

Vision-Language Navigation (VLN) requires embodied agents to navigate diverse 3D environments following natural language instructions [1–5]. To achieve robust performance, agents must combine accurate spatial perception with reliable decision-making strategies [6–9].

Early agents adopted sequence-to-sequence frameworks [1, 10], directly mapping language and visual observations into actions. Later works introduced map-based paradigms that explicitly encoded spatial connectivity via topological graphs [9, 11], incorporated semantic information for object-level reasoning [12–14], and leveraged grid-based [15, 16] or volumetric voxel-based representations [17] to capture 3D structure. For policy learning, agents evolved from pure imitation [18, 19] to hybrid approaches that combine imitation and reinforcement with tailored rewards [10, 20–23]. More recently, several agents have employed world models [24–26] to perform look-ahead planning. Despite these advances, existing agents typically ignore uncertainty in perception when making decisions. Their training recipes discourage expressing uncertainty or recognizing unreliable situations, instead incentivizing them to predict actions regardless of confidence [17]. For instance, as illustrated in Fig. 1 📍, agents may confuse visually similar doors, especially when the interior cues behind them provide insufficient evidence, leading to unreliable grounding of the correct target. Besides, Fig. 1 📍 illustrates how occlusions mask critical spatial information, introducing ambiguity in assessing path traversability. Previous agents often fail under such conditions, whereas uncertainty offers valuable cues about the reliability of perception and the feasibility of actions [11, 15, 17].

---

*Corresponding author

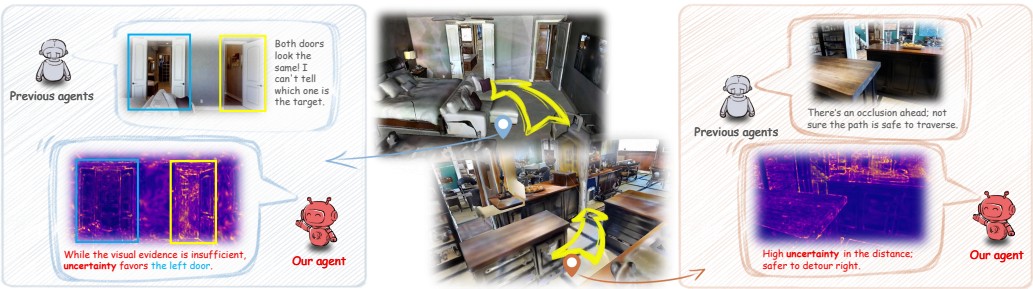

Figure 1: **Motivation.** Previous VLN agents typically ignore perceptual uncertainty when making decisions. As a result, they often confuse visually similar structures (*e.g.*, multiple doors) due to limited interior evidence (📍) and struggle when occlusions obscure spatial cues, leaving traversability ambiguous and causing unsafe or suboptimal paths (📍). In contrast, our agent explicitly models and leverages such uncertainty for more reliable navigation. Brighter colors indicate higher uncertainty.

In light of the foregoing discussions, this work explicitly models geometric, semantic, and appearance uncertainty in perception and consolidates them into a unified 3D Value Map for reliable navigation. **First**, our agent constructs a *Semantic Gaussian Map (SGM)* that represents the environment as a collection of differentiable 3D Gaussian primitives. Each primitive is initialized from sparse pseudo-lidar point clouds obtained from multi-view RGB-D observations and further enriched with semantic properties based on their object instance or stuff membership in the 3D scene. **Second**, building on SGM, the agent estimates three forms of perceptual uncertainty in navigation. *Geometric uncertainty* is modeled through variational inference, which approximates the posterior distribution over position and scale perturbations of Gaussians, thereby assessing structural reliability and enabling the pruning of unreliable primitives. In the same manner, *semantic uncertainty* is estimated by perturbing the semantic attributes of Gaussians, which reveals ambiguous interpretations and allows the agent to down-weight unreliable semantic cues during decision-making. *Appearance uncertainty* reflects the sensitivity of rendered observations to Gaussian-level perturbations, quantified by Fisher Information as the reconstruction loss surface curvature around each Gaussian. **Third**, our agent composes a unified *3D Value Map* by transforming these uncertainties into affordances and constraints within its perceptual space, thereby guiding informed and more reliable trajectories.

Our agent is evaluated on three VLN benchmarks, *i.e.*, R2R [1], RxR [27], and REVERIE [28] (§4.2, §4.3). It improves SR by **2**% and SPL by **1**% on R2R, yields **1.1**% and **1.7**% increases in SR and nDTW on RxR with comparable SDTW, and achieves **2.94**% and **2.57**% higher RGS and RGSPL scores on REVERIE. Extensive ablation studies confirm the contribution of each component (§4.4).

## 2 RELATED WORK

**Vision-Language Navigation (VLN).** Early VLN agents adopted sequence-to-sequence frameworks that directly map instructions and multi-view observations to actions [1, 10, 29]. Yet such models struggle with long-horizon reasoning and robustness in unseen environments, which has spurred a variety of extensions. As a primary step, subsequent agents introduced explicit memory mechanisms, such as topological graphs [11, 30] or episodic memory buffers [31–33], to better retain and recall spatial and semantic cues over extended trajectories. Later works further advanced the agent with transformer-based architectures that jointly encode instruction and observation [25, 34]. Moreover, extensive efforts are devoted to mitigating data limitations through instruction generation [35–39] and synthetic data creation [26]. Several works enable the agent to explore steps forward by anticipating future observations before decision-making [24, 25, 40]. To improve policy robustness, the combination of imitation learning [41] and reinforcement learning [42] has been widely adopted in VLN agents [20]. Others aim to reduce computational costs while maintaining VLN performance by designing lightweight cross-modal or selective memorization architectures [43]. Benefiting from the quality and speed of 3D Gaussian Splatting [44–46], a growing line of work has adopted it as the agent's scene representation, showing strong performance [5, 47–49].

In parallel, a few recent studies begin to explore uncertainty-related signals in VLN. For instance, VLN-Copilot [50] estimates **decision-level** uncertainty from the action distribution to decide when to request external large language model assistance. In contrast, our work focuses on **perception-**

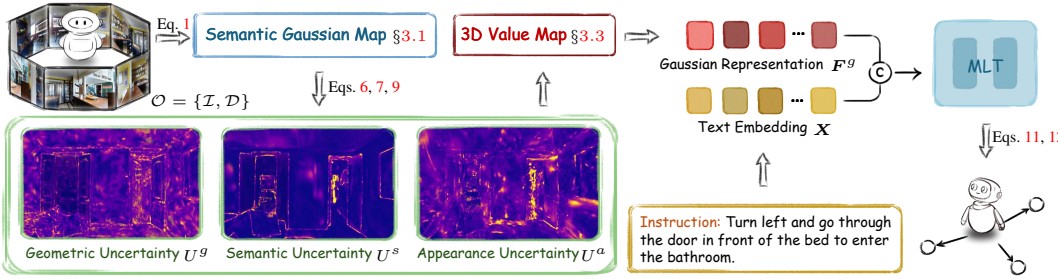

Figure 2: **Pipeline overview.** At each step, our agent constructs a *Semantic Gaussian Map (§3.1)* from its panoramic observation $\mathcal{O} = \{\mathcal{I}, \mathcal{D}\}$. On top of this map, it estimates geometric $U^g$, semantic $U^s$, and appearance $U^a$ uncertainties (§3.2) and embeds them back to obtain a unified *3D Value Map (§3.3)* that grounds affordances and constraints. Finally, Gaussian representations $F^g$ derived from the value map are concatenated with the instruction embedding $X$ and fed into a multi-layer transformer $\mathcal{F}^{\mathrm{MLT}}$ to predict the next action over candidate waypoints (§3.3).

**level** uncertainty that arises from the agent's observations and models geometric, semantic, and appearance reliability to support informed decision-making.

**Uncertainty Estimation in Deep Learning.** Uncertainty estimation has long been recognized as a central challenge in deep learning, with a variety of approaches proposed across vision and robotics [51]. A prominent line of work follows the Bayesian paradigm, which characterizes uncertainty through predictive distributions over model parameters, often approximated via variational inference, Laplace approximation, or sampling methods [52–54]. Another common strategy relies on ensembling, where multiple models trained with different initializations, data subsets, or hyperparameters are aggregated to approximate appearance uncertainty [55–59]. In parallel, sampling-based techniques like Hamiltonian Monte Carlo provide asymptotic guarantees but incur prohibitive costs for high-dimensional models [60]. To alleviate this computational burden, several works leverage regularization-based approximations, such as Monte Carlo Dropout [61] and its variants [62], which approximate Bayesian inference with minimal changes to standard training. Recent efforts exploit second-order information, where the Hessian or Fisher Information of the loss surface is approximated to assess how sensitive predictions are to parameter variations [63–65].

However, most existing approaches rely on implicit latent representations, where globally entangled features obscure uncertainty estimation and hinder region-specific reasoning. By contrast, the explicit structure of 3D Gaussian Splatting [44] provides a natural and interpretable way to associate physically meaningful attributes (*i.e.*, position, scale, semantics) with each primitive. While recent studies have explored this explicit structure for estimating uncertainty, they largely concentrate on novel view synthesis and image reconstruction [66, 67]. In contrast, our agent leverages these physically grounded primitives to construct a unified 3D Value Map, which explicitly quantifies uncertainty and encodes it as affordances and constraints to guide navigation.

# 3 METHOD

**Problem Formulation.** In VLN, an agent is placed in a 3D scene and required to reach a target location [1] (or identify a target object [28]) following instructions $\mathcal{X}$. At each step $t$, the agent receives a panoramic observation composed of multiple RGB views $\mathcal{I}_t = \{\mathcal{I}_{t,k} \in \mathbb{R}^{H \times W \times 3}\}_{k=1}^K$ and associated depth maps $\mathcal{D}_t = \{\mathcal{D}_{t,k} \in \mathbb{R}^{H \times W}\}_{k=1}^K$. Based on these observations, the agent learns a navigation policy $\pi(a_t | \mathcal{X}, \mathcal{I}_t, \mathcal{D}_t)$ that predicts actions $a_t \in \mathcal{A}_t$, which includes navigable neighbor nodes, previously observed nodes accessible via backtracking, and a [STOP] action.

**Overview (Fig. 2).** At each step, our agent constructs a *Semantic Gaussian Map (SGM)* from multi-view RGB-D observations, where primitives are enriched with semantic properties (§3.1). Building on SGM, the agent models *geometric*, *semantic*, and *appearance uncertainty* to capture the perceptual unreliability in VLN (§3.2). These uncertainties are then integrated into a *3D Value Map*, encoding affordances and constraints in the agent's perceptual space for decision-making (§3.3).

### 3.1 Semantic Gaussian Map

At each waypoint, the agent transforms multi-view RGB-D observations into a collection of differentiable 3D Gaussian primitives, each encoding both geometric and semantic properties. Through differentiable rendering, these primitives are jointly optimized to form a *Semantic Gaussian Map (SGM)*, which serves as the foundational substrate for subsequent uncertainty modeling.

**Initialization.** Given multi-view RGB-D observations $\mathcal{O}_t = \{\mathcal{I}_t, \mathcal{D}_t\}$ at step $t$, the agent first generates a sparse pseudo-lidar point cloud via camera-to-world transformation. Each pixel $(u, v)$ in $\mathcal{I}_{t,k}$ is back-projected into the 3D coordinates $(x, y, z)$ using its depth $D_{t,k}(u, v)$ and camera intrinsics $(c^u, c^v, f^x, f^y)$, where $(c^u, c^v)$ are the principal point and $(f^x, f^y)$ are the focal lengths:

$$z = D_{t,k}(u, v), \qquad x = \frac{(u - c^u)z}{f^x}, \qquad y = \frac{(v - c^v)z}{f^y}. \tag{1}$$

These 3D points are then transformed to world coordinates using the camera pose, yielding a sparse point cloud $\mathcal{P}_t$. Each point initializes a Gaussian primitive $g_i$, parameterized by position (mean) $\mu_i \in \mathbb{R}^3$, covariance matrix $\Sigma_i \in \mathbb{R}^{3\times3}$, opacity $\alpha_i \in [0, 1]$, spherical harmonics coefficients for color $c_i \in \mathbb{R}^3$, and semantic property $s_i \in \mathbb{R}^3$ ($t$ is omitted for simplicity). In detail, $\Sigma$ is factorized into a scale matrix $E$ and a rotation matrix $R$ as $\Sigma = REE^\top R^\top$, where $E = \mathrm{diag}(e^x, e^y, e^z)$ and $R$ is constructed from a unit quaternion $r \in \mathbb{R}^4$. For $s$, we apply SAM2 [68] to segment the panoramic observation $\mathcal{I}$ into spatially coherent regions $\{m_k\}_{k=1}^K$ and extract their CLIP [69] embeddings, which are then attached to the corresponding Gaussians as new semantic attributes.

**Construction.** SGM is progressively constructed by optimizing Gaussian primitives through differentiable rendering, which enforces consistency with the current observation. Specifically, the rendered color $\hat{\mathcal{I}}$ at pixel $(u, v)$ is obtained by $\alpha$-blending depth-ordered Gaussians:

$$\hat{I}(u, v) = \sum_i c_i \alpha_i' \prod_{j=1}^{i-1}(1 - \alpha_j') \in \mathbb{R}^3, \quad \alpha_i' = \alpha_i \cdot \exp\left(-\tfrac{1}{2}(x' - \mu_i')^\top \Sigma_i'^{-1}(x' - \mu_i')\right) \in \mathbb{R}^+, \tag{2}$$

where $x' = (u, v)$ and $\mu_i'$ is the Gaussian center in the image plane, and $\Sigma_i'$ is the 2D covariance.

Following the same principle, the rendered depth $\hat{D}(u, v)$ and semantic $\hat{S}(u, v)$ are computed as:

$$\hat{D}(u, v) = \sum_i z_i \alpha_i' \prod_{j=1}^{i-1}(1 - \alpha_j') \in \mathbb{R}^+, \quad \hat{S}(u, v) = \sum_i s_i \alpha_i' \prod_{j=1}^{i-1}(1 - \alpha_j') \in \mathbb{R}^3. \tag{3}$$

Furthermore, Gaussians with small scale often capture irrelevant surface noise, while low-opacity primitives represent negligible background clutter. These Gaussians contribute minimally to the agent's spatial comprehension and can potentially introduce misleading cues for its decision-making. Therefore, after several rounds of differentiable rendering optimization, we further refine SGM by retaining only Gaussians subject to the constraints $\|e_i\|_2 > \tau_e \wedge \alpha_i > \tau_\alpha$. Consequently, the refined SGM serves as a foundational substrate for subsequent uncertainty modeling and estimation.

### 3.2 Uncertainty Estimation

SGM provides an explicit 3D map enriched with spatial geometry and semantic context. On top of this map, three forms of perceptual uncertainty are modeled. Geometric uncertainty assesses structural reliability through perturbations of Gaussian position and scale. Semantic uncertainty exposes ambiguous interpretations at the object and region levels by perturbing semantic attributes. Appearance uncertainty characterizes inherent visual ambiguity in observations, arising from occlusions, texture inconsistencies, and other uncontrollable factors.

**Geometric Uncertainty.** To quantify spatial reliability of SGM, we model position and scale parameters of each Gaussian as random variables with learnable perturbations $\chi_i^\mu \in \mathbb{R}^3$ and $\chi_i^e \in \mathbb{R}^3$:

$$\mu_i' = \mu_i + \chi_i^\mu, \qquad e_i' = e_i + \chi_i^e, \tag{4}$$

where $\mu_i'$ and $e_i'$ denote perturbed spatial parameters that encode alternative structural hypotheses of $g_i$. The posterior distribution $p(\chi \,|\, \mathcal{O})$ over perturbations $\chi = \{\chi_i^\mu, \chi_i^e\}_i$ conditioned on observations $\mathcal{O}$ is generally intractable, as it involves integration over a high-dimensional continuous space. To approximate it, like [66], we introduce variational distributions $q_\phi(\chi) = \{q_{\phi_i^\mu}(\chi_i^\mu), q_{\phi_i^e}(\chi_i^e)\}_i$ and optimize them by minimizing the Kullback–Leibler (KL) divergence to true posterior $p(\chi|\mathcal{O})$:

$$\min_\phi d^{\mathrm{KL}}(q_\phi(\chi) \,\|\, p(\chi|\mathcal{O})) = \log p(\mathcal{O}) - \big( \underbrace{\mathbb{E}_{q_\phi(\chi)}[\log p(\mathcal{O}|\chi)] - d^{\mathrm{KL}}(q_\phi(\chi)\|p(\chi))}_{\text{Evidence Lower Bound (ELBO)}} \big). \tag{5}$$

Since $\log p(\mathcal{O})$ is constant with respect to $\boldsymbol{\chi}$, minimizing KL divergence is equivalent to maximizing ELBO, which serves as the training objective in learning $q_{\boldsymbol{\phi}}(\boldsymbol{\chi})$. In this process, the prior $p(\boldsymbol{\chi})$ is defined as a zero-mean Gaussian $\mathcal{N}(\mathbf{0}, \delta^2\mathbf{I})$ for position perturbations $\boldsymbol{\chi}^{\mu}$, and a scale-dependent uniform distribution $\mathcal{U}(-\eta\boldsymbol{e}, \eta\boldsymbol{e})$ for scale perturbations $\boldsymbol{\chi}^e$, where $\delta$ controls the standard deviation of position variances and $\eta$ determines the perturbation range relative to the original scale $\boldsymbol{e}$.

Based on the learned variational distribution $q_{\boldsymbol{\phi}}(\boldsymbol{\chi})$, the geometric uncertainty $U_i^g$ of each Gaussian $\boldsymbol{g}_i$ is estimated by the variability of its perturbations. In particular, we extract the standard deviations of position and scale perturbations from $q_{\boldsymbol{\phi}}$ and aggregate them into a scalar score as:

$$U_i^g = \|\mathcal{F}^{\text{std}}(q_{\boldsymbol{\phi}_i^{\mu}}(\boldsymbol{\chi}_i^{\mu}))\|_2 + \|\mathcal{F}^{\text{std}}(q_{\boldsymbol{\phi}_i^e}(\boldsymbol{\chi}_i^e))\|_2 \in \mathbb{R}^+, \tag{6}$$

where $\mathcal{F}^{\text{std}}(\cdot)$ is the operation that extracts the standard deviation of the variational distribution.

**Semantic Uncertainty.** In addition to geometric unreliability, agents also face semantic ambiguity, where object- and region-level understanding may be unstable. To capture this, we perturb the semantic attribute $\boldsymbol{s}_i$ of each Gaussian with a learnable offset $\boldsymbol{\chi}_i^s \in \mathbb{R}^3$, while keeping geometric parameters fixed to preserve spatial consistency. Following the same variational inference framework, we learn a posterior $q_{\boldsymbol{\phi}^s}(\boldsymbol{\chi}^s)$ by maximizing the corresponding ELBO, regularized by a zero-mean Gaussian prior $p(\boldsymbol{\chi}^s) = \mathcal{N}(\mathbf{0}, \epsilon^2\mathbf{I})$, where $\epsilon$ controls the perturbation magnitude. The semantic uncertainty $U_i^s$ of Gaussian $\boldsymbol{g}_i$ is defined as the variability of $\boldsymbol{\chi}_i^s$ under the posterior $q_{\boldsymbol{\phi}^s}(\boldsymbol{\chi}^s)$:

$$U_i^s = \|\mathcal{F}^{\text{std}}(q_{\boldsymbol{\phi}^s}(\boldsymbol{\chi}_i^s))\|_2 \in \mathbb{R}^+. \tag{7}$$

**Appearance Uncertainty.** To further capture visual instability, we define appearance uncertainty as the sensitivity of the reconstruction loss $\mathcal{L}^r = \frac{1}{2}\|\hat{\mathcal{I}} - \mathcal{I}\|_2^2$ to variations in SGM. In principle, such sensitivity is characterized by the Hessian matrix $\nabla_{\mathcal{G}}^2\mathcal{L}^r$ [64, 65]. Because computing this matrix directly is infeasible, like [63, 67], we adopt the Fisher Information as a tractable approximation:

$$\nabla_{\mathcal{G}}^2\mathcal{L}^r = \underbrace{\nabla_{\mathcal{G}}\hat{\mathcal{I}}\nabla_{\mathcal{G}}\hat{\mathcal{I}}^{\top}}_{\text{Fisher Information}} + \underbrace{(\hat{\mathcal{I}} - \mathcal{I})\nabla_{\mathcal{G}}^2\hat{\mathcal{I}}}_{\text{Residual Term}} \in \mathbb{R}^{(|\mathcal{G}| \cdot d^{\boldsymbol{g}}) \times (|\mathcal{G}| \cdot d^{\boldsymbol{g}})}, \tag{8}$$

where $\nabla_{\mathcal{G}}\hat{\mathcal{I}}$ denotes the gradient of the rendered observations with respect to all Gaussian parameters in $\mathcal{G}$, $\nabla_{\mathcal{G}}^2\hat{\mathcal{I}}$ represents their second-order derivatives, $|\mathcal{G}|$ is the number of Gaussians in SGM, and $d^{\boldsymbol{g}}$ is the feature dimension of each Gaussian. In a refined SGM, where $(\hat{\mathcal{I}} - \mathcal{I})$ in the Residual Term approaches zero, the Hessian reduces to the Fisher Information, which serves as a tractable proxy of the sensitivity. High Fisher Information reveals that even minor Gaussian shifts can induce large variations in the perceptual space, destabilizing both scene understanding and action predictions.

While Fisher Information avoids computing costly second-order derivatives, it still has the same dimension as the Hessian (*i.e.*, $(|\mathcal{G}| \cdot d^{\boldsymbol{g}}) \times (|\mathcal{G}| \cdot d^{\boldsymbol{g}})$), which remains computationally expensive. To reduce this cost, we group parameters associated with each Gaussian $\boldsymbol{g}_i \in \mathbb{R}^{d^{\boldsymbol{g}}}$, yielding a diagonal block of size $\mathbb{R}^{d^{\boldsymbol{g}} \times d^{\boldsymbol{g}}}$ within the Fisher Information matrix. Each block isolates the sensitivity of $\boldsymbol{g}_i$, quantifying the impact of its perturbations on the reconstruction loss. Based on this, the appearance uncertainty $U_i^a$ is defined as the log-determinant of the corresponding Fisher Information block:

$$U_i^a = \log\left|\nabla_{\boldsymbol{g}_i}\hat{\mathcal{I}}\nabla_{\boldsymbol{g}_i}\hat{\mathcal{I}}^{\top}\right| \in \mathbb{R}^+, \tag{9}$$

where $|\cdot|$ denotes the matrix determinant. The log-determinant quantifies the volume of the uncertainty ellipsoid in parameter space, yielding a scalar measure of the sensitivity for each Gaussian.

## 3.3 3D VALUE MAP

To operationalize the estimated uncertainties for navigation, we integrate them into a 3D Value Map. In traditional 3D scene reasoning and robotics, a value map represents a spatial field in which each element encodes task-relevant signals that guide downstream decisions, such as affordance fields [70], cost maps [71], and traversability [72] or reliability maps [73]. Following this notion, our 3D Value Map instantiates a value field on top of SGM, where each Gaussian is augmented with geometric, semantic, and appearance uncertainty estimates. These uncertainties provide unified reliability cues, which can be naturally interpreted as affordances and constraints for navigation.

**Construction.** By attaching $U_i^g$, $U_i^s$, and $U_i^a$ to each Gaussian $g_i$, we extend SGM into a 3D Value Map. For ease of notation, we reuse $g_i$ to denote the Gaussian representation of this value map:

$$g_i = \{ \boldsymbol{\mu}_i, \boldsymbol{e}_i, \boldsymbol{r}_i, \alpha_i, \boldsymbol{c}_i, \boldsymbol{s}_i, U_i^g, U_i^s, U_i^a \} \in \mathbb{R}^{20}. \tag{10}$$

This augmented representation preserves geometric and semantic information while further incorporating complementary uncertainty measures. Consequently, the 3D Value Map characterizes both the structural and semantic reliability of the environment, grounding affordances and constraints into the agent's observation space to support reliable decision-making.

**Action Prediction.** Given the instruction embedding $\boldsymbol{X} \in \mathbb{R}^{768}$, each $g_i$ is nonlinearly projected into a feature vector $\boldsymbol{F}^{g_i} \in \mathbb{R}^{768}$. This projection embeds all Gaussian attributes, including geometry, semantics, and uncertainty, into a unified feature space. In this space, the agent maintains a direct correspondence between local geometric structure and its associated uncertainty. $\boldsymbol{F}^{g_i}$ are then aggregated into a global representation. The aggregated representation $\boldsymbol{F}^g$ preserves the fine-grained coupling between geometry and uncertainty, enabling the agent to make decisions that are directly informed by such structure-aware uncertainty. Consequently, $\boldsymbol{F}^g$ is concatenated with $\boldsymbol{X}$ and processed by a multi-layer transformer $\mathcal{F}^{\text{MLT}}$ [11] to produce candidate node probabilities $\boldsymbol{p}$:

$$\boldsymbol{p} = \text{Softmax}\big(\mathcal{F}^{\text{MLT}}([\boldsymbol{F}^g, \boldsymbol{X}])\big) \in [0,1]^{|\mathcal{V}|}, \tag{11}$$

where $|\mathcal{V}|$ is the number of candidate waypoints and $[\cdot, \cdot]$ denotes concatenation. These scores are then aligned with the action space $\mathcal{A}$ through nearest-neighbor mapping $\mathcal{N}$:

$$\tilde{\boldsymbol{p}} = \mathcal{N}(\boldsymbol{p}, \mathcal{V}) \in [0,1]^{|\mathcal{V}|}, \tag{12}$$

where $\mathcal{N}$ denotes the mapping of node-level scores to executable actions via nearest-neighbor search. This fusion enables the agent to jointly reason about geometric structure and perceptual confidence, thereby promoting reliable and uncertainty-aware decision-making.

## 3.4 Loss Function

**SGM Loss.** To supervise SGM construction, we apply a pixel-wise rendering loss between the rendered outputs and ground-truth observations. Specifically, we combine $\mathcal{L}^1$ loss and Structural Similarity [74] loss $\mathcal{L}^{\text{SSIM}}$ for color consistency, and apply $\mathcal{L}^1$ for depth and semantic alignment:

$$\mathcal{L}^{\text{rgb}} = \|\hat{\mathcal{I}} - \mathcal{I}\|_1 + \mathcal{L}^{\text{SSIM}}(\hat{\mathcal{I}}, \mathcal{I}), \quad \mathcal{L}^{\text{depth}} = \|\hat{\mathcal{D}} - \mathcal{D}\|_1, \quad \mathcal{L}^{\text{sem}} = \|\hat{\mathcal{S}} - \mathcal{S}\|_1, \tag{13}$$

where $\mathcal{I}, \mathcal{D}, \mathcal{S}$ denote the ground-truth color, depth, and semantic features, respectively, while $\hat{\mathcal{I}}, \hat{\mathcal{D}}, \hat{\mathcal{S}}$ are the corresponding rendered outputs from current SGM.

**Navigation Loss.** Following the conventional procedure [11, 17, 30], our agent is optimized with a two-stage training scheme: pretraining with auxiliary objectives such as masked language modeling and single-step action prediction to strengthen multimodal representations, and finetuning with behavior cloning and pseudo-expert guidance to refine policy learning. (See details in **Appendix**.)

## 3.5 Implementation Details

**Topological Memory.** To support long-horizon reasoning, similar to [11, 17, 30], our agent maintains a dynamic topological memory that records both visited and navigable nodes as exploration unfolds. Each node is associated with multimodal features, including the 2D panoramic embeddings and the 3D Value Map representations, while edges encode traversability between locations. This memory forms a graph structure that evolves with the trajectory, enabling the agent to revisit prior viewpoints or evaluate alternative routes when needed. Regions that are ambiguous at the previous node may become more certain when viewed from a more informative location, while consistently uncertain areas remain marked as unreliable. By jointly storing 2D and 3D information in a spatially coherent manner, the memory provides global context that strengthens consistency and stabilizes decision-making in diverse environments. (See more details in **Appendix**.)

**Network Pretraining.** For R2R [1] and RxR [27], we adopt Masked Language Modeling (MLM) [75, 76] and Single-step Action Prediction (SAP) [21, 76] as auxiliary objectives. For REVERIE [28], we additionally employ Object Grounding (OG) [77] to enhance object-level reasoning. Pretraining is conducted for 100k iterations with a batch size of 64, optimized by Adam [78] with a learning rate of 1e-4. At each mini-batch, only one task is sampled with equal probability.

Table 1: **Quantitative results** on REVERIE [28]. '−': unavailable statistics. See §4.2 for more details.

| Method | REVERIE [28] | | | | | | | | | | | |
|---|---|---|---|---|---|---|---|---|---|---|---|---|
| | val unseen | | | | | | test unseen | | | | | |
| | TL ↓ | OSR ↑ | SR ↑ | SPL ↑ | RGS ↑ | RGSPL ↑ | TL ↓ | OSR ↑ | SR ↑ | SPL ↑ | RGS ↑ | RGSPL ↑ |
| RCM [20] | 11.98 | 14.23 | 9.29 | 6.97 | 4.89 | 3.89 | 10.60 | 11.68 | 7.84 | 6.67 | 3.67 | 3.14 |
| FAST-M [28] | 45.28 | 28.20 | 14.40 | 7.19 | 7.84 | 4.67 | 39.05 | 30.63 | 19.88 | 11.61 | 11.28 | 6.08 |
| RecBERT [21] | 16.78 | 35.02 | 30.67 | 24.90 | 18.77 | 15.27 | 15.86 | 32.91 | 29.61 | 23.99 | 16.50 | 13.51 |
| Airbert [81] | 18.71 | 34.51 | 27.89 | 21.88 | 18.23 | 14.18 | 17.91 | 34.20 | 30.28 | 23.61 | 16.83 | 13.28 |
| HAMT [76] | 14.08 | 36.84 | 32.95 | 30.20 | 18.92 | 17.28 | 13.62 | 33.41 | 30.40 | 26.67 | 14.88 | 13.08 |
| HOP [82] | 16.46 | 36.24 | 31.78 | 26.11 | 18.85 | 15.73 | 16.38 | 33.06 | 30.17 | 24.34 | 17.69 | 14.34 |
| DUET [11] | 22.11 | 51.07 | 46.98 | 33.73 | 32.15 | 23.03 | 21.30 | 56.91 | 52.51 | 36.06 | 31.88 | 22.06 |
| DUET-Imagine [40] | − | − | 48.28 | 33.76 | 32.97 | 23.25 | − | − | − | − | − | − |
| COSMO [43] | − | 56.09 | 50.81 | 35.93 | − | − | − | 59.33 | 52.53 | 36.12 | − | − |
| GridMM [16] | 23.20 | 57.48 | 51.37 | 36.47 | 34.57 | 24.56 | 19.97 | 59.55 | 53.13 | 36.60 | 34.87 | 23.45 |
| LANA [83] | 23.18 | 52.97 | 48.31 | 33.86 | 32.86 | 22.77 | 18.83 | 57.20 | 51.72 | 36.45 | 32.95 | 22.85 |
| BEVBert [15] | − | 56.40 | 51.78 | 36.37 | 34.71 | 24.44 | − | 57.26 | 52.81 | 36.41 | 32.06 | 22.09 |
| BSG [30] | 24.71 | 58.05 | 52.12 | 35.59 | 35.36 | 24.24 | 22.90 | 62.83 | 56.45 | 38.70 | 33.15 | 22.34 |
| VER [17] | 23.03 | 61.09 | 55.98 | 39.66 | 33.71 | 23.70 | 24.74 | 62.22 | 56.82 | 38.76 | 33.88 | 23.19 |
| 3DGS-VLN [5] | 22.22 | 58.81 | 53.59 | 37.67 | 36.73 | 26.74 | 20.05 | 56.93 | 52.93 | 36.93 | 35.65 | 25.76 |
| **Ours** | 22.38±0.14 | **61.98**±0.21 | **56.37**±0.19 | 37.64±0.24 | **37.65**±0.16 | **27.01**±0.20 | 20.14±0.11 | 60.12±0.23 | 55.90±0.20 | **38.77**±0.26 | **35.68**±0.17 | 25.50±0.18 |

**Network Finetuning.** Following standard protocol [11], we finetune the pretrained model using DAgger [79]. For REVERIE [28], an additional Object Grounding (OG) loss is incorporated with a weight of 0.20. Finetuning is performed for 25k iterations with a batch size of 8 and a learning rate of 1e-5. The best checkpoint is chosen based on performance of `val unseen` split.

**Testing.** At each navigable viewpoint, our agent constructs a SGM from panoramic observations and extends it into a 3D Value Map for reliable action prediction. This process terminates once the agent reaches the target location or decides to execute the [STOP] action. (See details in **Appendix**.)

**Runtime Analysis.** The main overhead arises from constructing the 3D Value Map, particularly semantic attribute extraction in SGM and uncertainty estimation. For training, we mitigate this cost through offline pretraining. During inference, RGB-D observations are resized to $224 \times 224$, and SAM2 [68] can be flexibly replaced by lightweight variants to trade off segmentation quality against runtime efficiency. Once the 3D Value Map is established, action prediction incurs negligible additional cost compared to existing VLN agents [11]. (See more details in **Appendix**.)

## 4 EXPERIMENT

### 4.1 EXPERIMENTAL SETUP

**Datasets.** We evaluate our agent on three benchmarks, each posing distinct challenges for VLN. All datasets are built upon the Matterport3D simulator [80], and are split into `train`, `val-seen`, `val-unseen`, and `test` sets according to scenes. **REVERIE** [28] provides 21,702 high-level instructions paired with 4,140 remote target objects. The agent must navigate to the described region and precisely ground the referred object. **R2R** [1] contains 7,189 shortest-path trajectories from 90 indoor scenes with 22K instructions, where the agent is required to follow detailed step-by-step directions. **RxR** [27] offers 126K multilingual instructions (*i.e.*, English, Hindi, Telugu) over 16,522 trajectories, requiring the agent to cope with long-horizon navigation across diverse languages.

**Evaluation Metrics.** We comprehensively evaluate agents using standard metrics [11] across different benchmarks. For R2R [1], we report Success Rate (SR), Trajectory Length (TL), Navigation Error (NE), Oracle Success Rate (OSR), and Success weighted by Path Length (SPL). For RxR [27], we additionally adopt Normalized Dynamic Time Warping (nDTW) and Success weighted nDTW (SDTW) to assess trajectory fidelity and path alignment. For REVERIE [28], evaluation further considers Remote Grounding Success (RGS) and RGS weighted by Path Length (RGSPL), which measure whether the agent successfully localizes the target object at the correct location.

### 4.2 QUANTITATIVE COMPARISON RESULT

Our results are averaged over five runs on three datasets, with standard deviations reported.

**Performance on REVERIE [28].** Table 1 reports the results on REVERIE, which evaluates the agent's ability to ground remote target objects given high-level instructions. On the `val unseen` split, our agent outperforms the best reported results (*i.e.*, BEVBert [15]) by a significant margin in terms of RGS (**37.65**% *vs* 34.71%) and RGSPL (**27.01**% *vs* 24.44%). These improvements of **3.94**% in RGS and **3.31**% in RGSPL clearly demonstrate the effectiveness of our 3D Value Map for accurate navigation and precise object grounding.

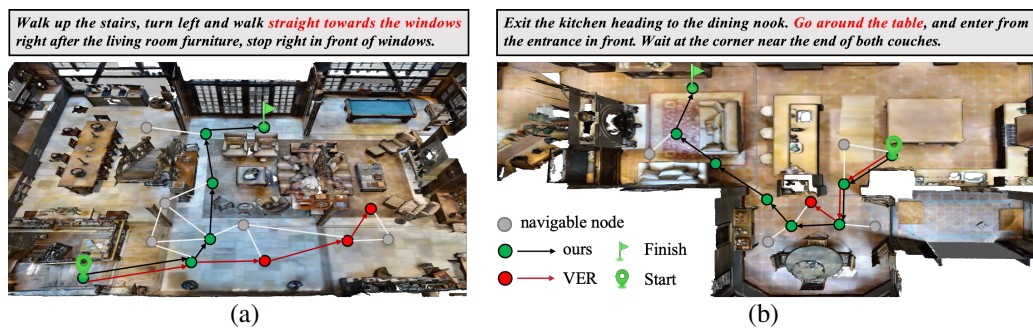

(a)                                                   (b)

Figure 3: **Qualitative results** on R2R [1]. (a) Under the instruction *"straight towards the windows"*, VER [17] misinterprets the layout and stops early, whereas our agent correctly follows the path and reaches the landmark. (b) Our agent bypasses the obstacle and enters the designated region, while VER halts at the *"table"* without completing the task. See §4.3 for more details.

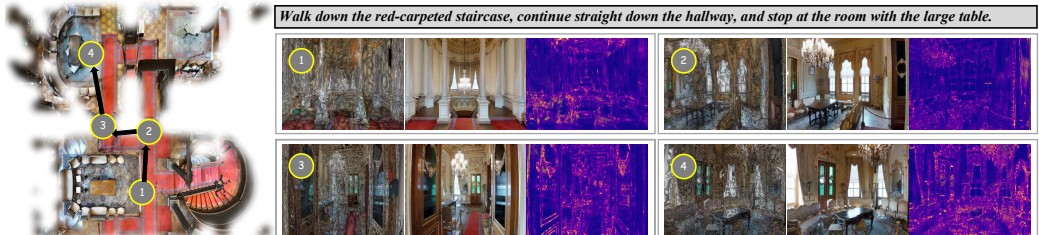

Figure 4: **Representative visual results** on R2R [1]. At each step, we show the constructed SGM, the rendered observations, and the aggregated uncertainty map. While SGM captures the geometry and semantic layout, the uncertainty emphasizes ambiguous regions such as reflective surfaces and repetitive structures, offering complementary cues for reliable grounding. See §4.3 for more details.

**Performance on R2R [1].** As shown in Table 2, our agent consistently surpasses recent state-of-the-art methods on R2R. On the `val unseen` split, it achieves an SR of **78**% compared to 76% from VER [17] and improves SPL from 65% to **66**%, corresponding to gains of **2**% in SR and **1**% in SPL. These results clearly highlight the ability of our agent to follow detailed instructions in unseen environments.

Table 2: **Quantitative results** on R2R [1] `val unseen`. '−': unavailable statistics. See §4.2 for more details.

| Method | R2R [1] | | | | | | | |
|---|---|---|---|---|---|---|---|---|
| | *val unseen* | | | | *test unseen* | | | |
| | TL ↓ | NE ↓ | SR ↑ | SPL ↑ | TL ↓ | NE ↓ | SR ↑ | SPL ↑ |
| HAMT [76] | 11.46 | 2.29 | 66 | 61 | 12.27 | 3.93 | 65 | 60 |
| DUET [11] | 13.94 | 3.31 | 72 | 60 | 14.73 | 3.65 | 69 | 59 |
| DUET-Imagine [40] | 14.35 | 3.19 | 72 | 60 | 15.35 | 3.52 | 71 | 60 |
| COSMO [43] | – | 3.15 | 73 | 61 | – | 3.43 | 71 | 58 |
| LANA [83] | 12.0 | – | 68 | 62 | 12.6 | – | 65 | 60 |
| GridMM [16] | 13.27 | 2.83 | 75 | 64 | 14.43 | 3.35 | 73 | 62 |
| BEVBert [15] | 14.55 | 2.81 | 75 | 64 | – | 3.13 | 73 | 62 |
| BSG [30] | 14.90 | 2.89 | 74 | 62 | 14.86 | 3.19 | 73 | 62 |
| VER [17] | 14.83 | 2.80 | 76 | 65 | 15.23 | 2.74 | 76 | 66 |
| 3DGS-VLN [5] | 14.83 | 2.43 | 77 | 66 | 14.58 | 3.17 | 75 | 65 |
| **Ours** | 14.79±0.12 | 2.12±0.15 | **78**±0.13 | **66**±0.17 | 14.68±0.14 | 3.17±0.06 | **76**±0.21 | **66**±0.29 |

**Performance on RxR [27].** Table 3 presents the results on RxR, which features longer paths and multilingual instructions. Our agent attains higher SR and nDTW (**65.2**% *vs* 64.1%, **65.6**% *vs* 63.9%) and comparable SDTW (**53.5**% *vs* 52.6%) on the `val unseen` split. Such improvements further demonstrate the benefit of the uncertainty information in long-horizon navigation.

Table 3: **Quantitative results** on RxR [27] `val unseen`. '−': unavailable statistics. See §4.2.

| Method | NE ↓ | SR ↑ | nDTW ↑ | SDTW ↑ |
|---|---|---|---|---|
| LSTM [27] | 10.9 | 22.8 | 38.9 | 18.2 |
| EnvDrop+ [84] | – | 42.6 | 55.7 | – |
| HAMT [76] | – | 56.5 | 63.1 | 48.3 |
| EnvEdit [85] | – | 62.8 | 68.5 | 54.6 |
| BEVBert [15] | 4.6 | 64.1 | 63.9 | 52.6 |
| **Ours** | **4.2**±0.08 | **65.2**±0.22 | **65.6**±0.19 | **53.5**±0.17 |

## 4.3  QUALITATIVE COMPARISON RESULT

**Case Studies.** We compare our agent with VER [17] on the R2R `val unseen` split. In Fig. 3(a), with multiple visually similar *"windows"*, VER misgrounds the target and deviates early, while our agent resolves the ambiguity and follows the correct landmarks. In Fig. 3(b), the instruction requires bypassing the *"table"* and reaching *"the corner near the couches"*. VER collides with the table and stops, whereas our agent detours safely and completes the instruction. These cases show how uncertainty helps disambiguate confounding structures and encode traversability constraints.

Moreover, Fig. 4 presents a step-wise visualization of the agent's trajectory. At each step, we show the constructed SGM, the rendered observations, and the aggregated uncertainty map (summing ge-

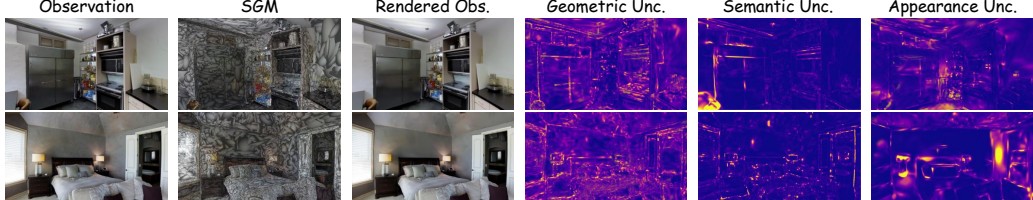

Figure 5: **Visualization** of diverse perceptual forms. From left to right: current observation, SGM, rendered observation, geometric uncertainty map, semantic uncertainty map, appearance uncertainty map. Brighter colors indicate higher uncertainty. See §4.3 for more details.

ometric, semantic, and appearance components). SGM captures the geometry and semantic layout, while the uncertainty highlights visually ambiguous regions such as reflective surfaces, chandeliers, and repetitive structures. These complementary views demonstrate how our agent grounds the instruction throughout navigation and show that uncertainty provides additional information.

**Visualization.** Fig. 5 illustrates our diverse perceptual forms. *i*) SGM preserves detailed geometric structures while maintaining high-fidelity rendering of the scene. *ii*) Geometric uncertainty reveals structural reliability, particularly highlighting uncertain boundaries and irregular surfaces. *iii*) Semantic uncertainty exposes ambiguity in object- and region-level interpretations, reflecting unstable semantic cues. *iv*) Appearance uncertainty highlights regions where rendered observations are highly sensitive to visual variations, *e.g.*, texture complexity, occlusions, or lighting variations.

## 4.4 DIAGNOSTIC EXPERIMENT

For thorough examination, we conduct a series of ablative studies on the `val unseen` split of R2R [1] and REVERIE [28].

**Key Component Analysis.** We first study the efficacy of the core components of our framework, *i.e.*, SGM (§3.1) and 3D Value Map (3DVM, §3.3). In Table 4, row #1 gives the performance of our base agent DUET [11]. For row #2, the scores are obtained by using SGM as the 3D scene representation without uncertainty values. In contrast, row #3 leverages only the uncertainty information (*i.e.*, $U^g$, $U^s$, $U^a$) as the 3D scene representation, without the raw Gaussian parameters. Row #4 reports the scores of our full framework. *i*) Row #1 *vs* #2: SGM leads to notable performance improvements against the baseline (*e.g.*, $32.15\% \rightarrow$ **35.48**% RGS on REVERIE). This demonstrates that the agent

Table 4: **Ablation studies** on `val unseen` split of R2R [1] and REVERIE [28]. See §4.4 for more details.

| # | Components | | R2R [1] | | REVERIE [28] | | |
|---|---|---|---|---|---|---|---|
| | SGM | 3DVM | SR ↑ | SPL ↑ | SR ↑ | RGS ↑ | RGSPL ↑ |
| 1 | – | – | 72.22 | 60.41 | 46.98 | 32.15 | 23.03 |
| 2 | ✓ | – | 76.21 | 64.57 | 50.20 | 35.48 | 25.64 |
| 3 | – | ✓ | 74.20 | 62.89 | 49.12 | 34.02 | 24.71 |
| 4 | ✓ | ✓ | **78.32** | **66.47** | **53.37** | **37.65** | **27.01** |

benefits from the geometric structure and semantic cues within SGM, achieving stronger navigation performance. *ii*) Row #1 *vs* #3: the uncertainty information boosts the performance of the baseline (*e.g.*, $72.22\% \rightarrow$ **74.20**% SR on R2R), which indicates that perceptual uncertainty inherent in navigation encodes informative cues that assist navigation decisions. *iii*) Row #2 *vs* #3: Explicit 3D structure with contextual awareness provides a stronger foundation for navigation than uncertainty alone (*e.g.*, **35.48**% *vs* 34.02% RGS on REVERIE). *iv*) Row #1 *vs* #4: Combining all contributions results in the largest gain over baseline, which confirms the effectiveness of our overall design.

**Analysis on SGM (§3.1).** We investigate how the scale of SGM (*i.e.*, the number of Gaussians) affects navigation performance. To control SGM scale, we apply pruning thresholds $\tau_e$ and $\tau_\alpha$ to filter out Gaussians with small scale

Table 5: **Effectiveness** of $\tau_e$ and $\tau_\alpha$ on `val unseen` splits of R2R [1] and REVERIE [28]. $N^{|g|}$ are Gaussian count within SGM.

| # | Pruning | | $N^{|g|}$ ↓ | FPS ↑ | R2R [1] | | REVERIE [28] | | |
|---|---|---|---|---|---|---|---|---|---|
| | $\tau_e$ | $\tau_\alpha$ | | | SR ↑ | SPL ↑ | SR ↑ | RGS ↑ | RGSPL ↑ |
| 1 | 0.00 | 0.000 | 50,000 | 11.2 | 77.30 | 63.26 | 52.00 | 35.00 | 26.50 |
| 2 | 0.01 | 0.002 | 45,000 | 13.1 | 77.87 | 64.80 | 52.70 | 35.30 | 27.00 |
| 3 | **0.015** | **0.005** | **42,000** | **15.5** | **78.32** | **66.47** | **53.37** | **37.65** | **27.01** |
| 4 | 0.02 | 0.010 | 35,000 | 18.7 | 74.80 | 61.68 | 46.50 | 32.30 | 24.80 |

($\|e_i\|_2 < \tau_e$) or low opacity ($\alpha_i < \tau_\alpha$), as these typically represent noise or irrelevant background clutter. Table 5 shows that, *i*) Slightly removing low-contribution Gaussians improves action accuracy (Row #2). *ii*) Moderate additional pruning yields clear rendering speedups while maintaining competitive accuracy (Row #3). *iii*) Aggressive removal markedly degrades performance (Row #4).

**Analysis on 3D Value Map (§3.3).** In Table 6, we investigate the contribution of different uncertainty types in our 3D Value Map. Row #1 utilizes SGM as the 3D scene representation. *i*) Row #1 *vs* (#2 or #3): Consistent performance gains appear when incorporating any form of perceptual uncertainty, con-

Table 6: **Effectiveness of** $U^g$, $U^s$, $U^a$ on `val unseen` of R2R [1] and REVERIE [28]. See §4.4 for more details.

| # | Uncertainty | | | R2R [1] | | REVERIE [28] | | |
|---|---|---|---|---|---|---|---|---|
| | $U^g$ | $U^s$ | $U^a$ | SR ↑ | SPL ↑ | SR ↑ | RGS ↑ | RGSPL ↑ |
| 1 | – | – | – | 76.21 | 64.57 | 50.20 | 35.48 | 25.64 |
| 2 | ✓ | ✓ | – | 77.05 | 65.12 | 51.82 | 36.96 | 26.11 |
| 3 | – | – | ✓ | 76.86 | 65.31 | 50.94 | 35.68 | 26.02 |
| 4 | ✓ | ✓ | ✓ | **78.32** | **66.47** | **53.37** | **37.65** | **27.01** |

firming that such signals provide useful guidance for navigation. *ii*) Row #2 *vs* #3: Geometric and semantic uncertainty contribute richer navigational cues than appearance uncertainty, as the agent benefits more from recognizing uncertain spatial structure or semantic interpretation than from sensitivity in visual rendering. *iii*) Configuration with all uncertainties achieves the best performance, highlighting their complementary roles.

## 5 CONCLUSION

This work presents a framework for Vision-Language Navigation that explicitly models geometric, semantic, and appearance uncertainty on top of a Semantic Gaussian Map. By integrating these uncertainties into a unified 3D Value Map, our agent grounds affordances and constraints into its perceptual space and achieves more reliable decision-making. Experiments across R2R, RxR, and REVERIE demonstrate consistent improvements over strong baselines, while qualitative analyses further validate the effectiveness of our uncertainty-aware design.

## ACKNOWLEDGEMENT

This work was supported by Zhejiang Provincial Natural Science Foundation of China (No. LR26F020002), Fundamental Research Funds for the Central Universities (226-2025-00057), and the State Key Laboratory of Brain Cognition and Braininspired Intelligence Technology (No. SKLBI-K2025004).

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

## SUMMARY OF THE APPENDIX

This appendix contains additional details for the ICLR 2026 submission, titled *Uncertainty-Aware Gaussian Map for Vision-Language Navigation*. The appendix is organized as follows:

- §A summarizes the notations used throughout the framework.
- §B reports additional model details.
- §C gives more runtime analysis.
- §D covers additional experiments.
- §E offers a discussion of our uncertainties and failure cases.
- §F provides a discussion of the limitations and future, societal impact, terms of use, privacy, and license, and use of large language models.

## A LIST OF SYMBOLS.

Table 7 concisely lists the symbols, excluding unnecessary subscripts for clarity.

| Notation | Description | Index |
|---|---|---|
| $\mathcal{X}$ | Natural language instructions | §3 |
| $\mathcal{I}_t$ | RGB images at step $t$ | §3.1; Eq. (2)&(3)&(4)&(13) |
| $\mathcal{D}_t$ | Depth images at step $t$ | §3.1; Eq. (2)&(3)&(6)&(13) |
| $\mathcal{O}_t$ | Multi-view RGB-D observations at step $t$ | §3.1 |
| $\mathcal{P}_t$ | Sparse point cloud from observations | §3.1; Eq. (1) |
| $\mathcal{A}_t$ | Predicted action at step $t$ | §3 |
| $\boldsymbol{\mu}_i$ | Position (mean) of Gaussian primitive $i$ | §3.1 |
| $\boldsymbol{\Sigma}_i$ | Covariance matrix of Gaussian primitive $i$ | §3.1 |
| $\alpha_i$ | Opacity of Gaussian primitive $i$ | §3.1; Eq. (2)&(3)&(4) |
| $\boldsymbol{c}_i$ | Color (spherical harmonics) of Gaussian primitive $i$ | §3.1; Eq. (2) |
| $\boldsymbol{s}_i$ | Semantic property of Gaussian primitive $i$ | §3.1; Eq. (3)&(7) |
| $\boldsymbol{E}_i$ | Scale matrix of Gaussian primitive $i$ | §3.1 |
| $\boldsymbol{R}_i$ | Rotation matrix of Gaussian primitive $i$ | §3.1 |
| $\boldsymbol{r}_i$ | Unit quaternion for rotation of Gaussian primitive $i$ | §3.1 |
| $\boldsymbol{g}_i$ | Gaussian primitive $i$ representation | §3.1; Eq. (2)&(3)&(4) |
| $\hat{\mathcal{I}}$ | Rendered RGB image | §3.1; Eq. (2)&(8)&(13) |
| $\hat{\mathcal{D}}$ | Rendered depth map | §3.1; Eq. (3)&(13) |
| $\hat{F}^\sigma$ | Rendered semantic feature | §3.1; Eq. (3)&(13) |
| $\boldsymbol{\chi}_i^\mu$ | Perturbation for position of Gaussian $i$ | §3.2; Eq. (4) |
| $\boldsymbol{\chi}_i^e$ | Perturbation for scale of Gaussian $i$ | §3.2; Eq. (4) |
| $\boldsymbol{\chi}_i^s$ | Perturbation for semantic of Gaussian $i$ | §3.2; Eq. (7) |
| $q_\phi(\boldsymbol{\chi})$ | Variational distribution for perturbations | §3.2; Eq. (5) |
| $q_{\phi_i^\mu}(\boldsymbol{\chi}_i^\mu)$ | Variational distribution for position perturbation of Gaussian $i$ | §3.2; Eq. (6) |
| $q_{\phi_i^e}(\boldsymbol{\chi}_i^e)$ | Variational distribution for scale perturbation of Gaussian $i$ | §3.2; Eq. (6) |
| $U_i^g$ | Geometric uncertainty of Gaussian $i$ | §3.2; Eq. (6) |
| $U_i^s$ | Semantic uncertainty of Gaussian $i$ | §3.2; Eq. (7) |
| $U_i^a$ | Appearance uncertainty of Gaussian $i$ | §3.2; Eq. (9) |
| $\boldsymbol{X}$ | Instruction embedding | §3.3; Eq. (11)&(12) |
| $\boldsymbol{F}^{g_i}$ | Projected feature of Gaussian $i$ | §3.3; Eq. (11) |
| $\boldsymbol{F}^g$ | Aggregated Gaussian representation | §3.3; Eq. (11) |
| $\boldsymbol{p}$ | Candidate node probabilities | §3.3; Eq. (11) |
| $\tilde{\boldsymbol{p}}$ | Action probabilities after mapping | §3.3; Eq. (12) |
| $\mathcal{L}^{\text{rgb}}$ | RGB rendering loss | §3.4; Eq. (13) |
| $\mathcal{L}^{\text{depth}}$ | Depth rendering loss | §3.4; Eq. (13) |
| $\mathcal{L}^{\text{sem}}$ | Semantic rendering loss | §3.4; Eq. (13) |

† Subscript $t$ denotes the navigation step.

Table 7: Notation and Description of Key Symbols.

# B MODEL DETAILS

Our approach is built upon the 2D observation–based baseline DUET [11], and the proposed 3D Value Map serves as an additional decision branch on top of it. In this part, we complement the details by introducing: **i) 2D Action Score** and **ii) Navigation Losses**.

## B.1 2D ACTION SCORE

Besides the 3D Value Map branch, we retain the 2D perception pathway inherited from DUET [11], which leverages 2D panoramic observations to guide navigation. The panoramic views and detected objects are first encoded by a multi-layer transformer (MLT) into 2D visual embeddings $\boldsymbol{F}^{\text{2D}} \in \mathbb{R}^{768}$. These features are then concatenated with the instruction embedding $\boldsymbol{X} \in \mathbb{R}^{768}$, and passed through another transformer head $\mathcal{F}^{\text{MLT}}$ to yield 2D action scores:

$$\boldsymbol{p}^{\text{2D}} = \text{Softmax}(\mathcal{F}^{\text{MLT}}([\boldsymbol{F}^{\text{2D}}, \boldsymbol{X}])) \in [0, 1]^{|\mathcal{V}|}, \tag{14}$$

where $[,]$ denotes concatenation and $|\mathcal{V}|$ is the number of candidate viewpoints. Next, we apply a nearest-neighbor function $\mathcal{N}$ to aggregate $\boldsymbol{p}^{\text{2D}}$ across neighboring nodes $\mathcal{V}$ in topological memory:

$$\boldsymbol{p}^{\hat{\text{2D}}} = \mathcal{N}(\boldsymbol{p}^{\text{2D}}, \mathcal{V}) \in [0, 1]^{|\mathcal{V}|}. \tag{15}$$

This operation merges scores from spatially adjacent nodes and outputs a unified value for each candidate, thereby aligning the predictions with the action space $\mathcal{A}$.

## B.2 NAVIGATION LOSSES

Following standard protocol [11, 17], our training follows a two-stage paradigm: pretraining with auxiliary objectives to enhance multimodal representations, and fine-tuning with behavior cloning and pseudo-expert guidance to refine navigation policy.

In the pretraining stage, three objectives are used depending on the benchmark. For R2R [1] and RxR [27], we include Masked Language Modeling (MLM) and Single-step Action Prediction (SAP) tasks. For REVERIE [28], we further incorporate Object Grounding (OG) to support precise localization of target objects. The corresponding losses are formulated as:

$$\mathcal{L}^{\text{MLM}} = -\log p(w_i|\mathcal{X}_{\setminus i}, \mathcal{R}), \tag{16}$$

$$\mathcal{L}^{\text{SAP}} = \sum_{t=1}^{T} -\log p(a_t^*|\mathcal{X}, \mathcal{R}_{<t}), \tag{17}$$

$$\mathcal{L}^{\text{OG}} = -\log p(o^*|\mathcal{X}, \mathcal{R}), \tag{18}$$

where $\mathcal{X}$ is the instruction sequence, $w_i$ a randomly masked token, and $\mathcal{X}_{\setminus i}$ its remaining context. $\mathcal{R}$ denotes the trajectory, with $\mathcal{R}_{<t}$ indicating the path prefix. The expert action at step $t$ is $a_t^*$, and $o^*$ represents the target object.

During fine-tuning, we adopt DAgger [11, 79], which alternates between agent rollouts and pseudo-expert corrections. The pseudo-expert leverages the partially constructed topological memory to generate shortest-path guidance, allowing the agent to recover from suboptimal actions and gradually improve its policy in unseen environments.

# C  RUNTIME ANALYSIS

Table 8: **Runtime Analysis** on R2R [1] `val unseen`. Our feature time is decomposed into SGM, variational inference (VI), and Fisher Information (FI) estimation. Wall-clock Time measures per-step latency, Mem. is peak GPU memory usage, and FLOPs measure computational complexity

| Method | Wall-clock Time (s) ↓ | | | | | | Mem. (GB) ↓ | FLOPs (G) ↓ | | | R2R | |
|---|---|---|---|---|---|---|---|---|---|---|---|---|
| | SGM | VI | FI | Feature | Action | Total | | Feature | Action | Total | SR ↑ | SPL ↑ |
| BEVBert [15] | – | – | – | 2.13 | 2.41 | 4.54 | 4.73 | 15.71 | 25.66 | 41.37 | 75 | 64 |
| DUET [11] | – | – | – | 0.42 | 0.89 | 1.31 | 3.13 | 2.71 | 18.68 | 21.39 | 72 | 60 |
| **Ours (MobileSAM)** | 0.62 | 0.11 | 0.19 | 0.92 | 1.71 | 2.63 | 3.45 | 3.82 | 20.53 | 24.35 | 76 | 65 |
| **Ours** | 1.47 | 0.11 | 0.19 | 1.77 | 1.71 | 3.48 | 3.84 | 4.36 | 20.53 | 24.89 | 78 | 66 |

## C.1  RUNTIME ANALYSIS ON THE OVERALL DESIGN

Table 8 reports the runtime decomposition in terms of wall-clock time, memory usage, and FLOPs across different components on R2R [1] `val unseen` split in inference. Compared to DUET [11], our agent achieves substantial performance gains (**+6%** SR) with only modest increases in runtime (**+1.35**s), memory (**+0.71**GB) and FLOPs (**+2.96**G), highlighting the efficiency of our design.

## C.2  RUNTIME ANALYSIS ON SGM

As shown in Table. 8, the majority of overhead arises from constructing SGM, dominated by semantic attribute extraction with SAM2 [68] (**1.47**s). Owing to the flexibility of our framework, SAM2 can be seamlessly replaced with lightweight variants (*e.g.*, MobileSAM [86]), enabling flexible trade-offs between segmentation quality and runtime efficiency (*e.g.*, **78**% → 76% SR with 1.47s→**0.62**s). This flexibility allows our agent to adapt to different deployment scenarios.

## C.3  RUNTIME ANALYSIS ON VARIATIONAL INFERENCE

Table 9: **Runtime Analysis** of Variational Inference (VI) on R2R [1] `val unseen`.

| Perturbed Parameters | Inference Time (s) ↓ | SR ↑ | SPL ↑ |
|---|---|---|---|
| None (*w/o* VI) | – | 72 | 60 |
| Position + Scale | 0.08 | 77 | 65 |
| Semantic Only | 0.07 | 76 | 64 |
| All (Pos.+Scale+Sem.) | 0.11 | **78** | **66** |

As illustrated in Table. 8, VI introduces only minimal cost (**0.11**s), since it perturbs Gaussian parameters with lightweight noise and updates variational distributions. In addition, to further assess the efficiency of VI, we measure per-step inference time and performance on R2R [1] `val unseen` split in Table 9. We can observe that perturbing only spatial parameters (*i.e.*, position and scale) or only semantic attributes incurs negligible overhead (**0.08**s and **0.07**s, respectively). When applied jointly, VI maintains a similarly low cost (**0.11**s) while yielding the best navigation performance.

## C.4  RUNTIME ANALYSIS ON FISHER INFORMATION ESTIMATION

Fisher Information (FI) estimation emerges as a lightweight component in our framework, requiring only **0.19**s per step (Table 8). This efficiency stems from approximating FI as the outer product of first-order gradients, which circumvents the costly computation of the full Hessian. Furthermore, we adopt a block-diagonal approximation at the Gaussian level, isolating sensitivity within each primitive and further reducing complexity.

## D ADDITIONAL EXPERIMENTS

This section presents supplementary experiments, including hyperparameter sensitivity analysis and statistical significance tests.

### D.1 HYPERPARAMETER EXPERIMENTS

Table 10: **Sensitivity Analysis** of uncertainty-related hyperparameters on R2R `val unseen` split. (a) $\delta$ and (b) $\eta$ regulate geometric uncertainty, while (c) $\varepsilon$ governs semantic uncertainty.

(a) Sensitivity to $\delta$

| $\delta$ | SR ↑ | SPL ↑ |
|---|---|---|
| 0.0015 | 78.27 | 66.41 |
| 0.002 | 78.29 | 66.44 |
| **0.0025** | **78.32** | 66.47 |
| 0.003 | 78.30 | 66.46 |
| 0.0035 | 78.28 | **66.48** |
| 0.005 | 78.25 | 66.40 |

(b) Sensitivity to $\eta$

| $\eta$ | SR ↑ | SPL ↑ |
|---|---|---|
| 0.05 | 78.30 | 66.45 |
| **0.1** | **78.32** | **66.47** |
| 0.15 | 78.30 | 66.44 |
| 0.2 | 78.28 | 66.41 |

(c) Sensitivity to $\varepsilon$

| $\varepsilon$ | SR ↑ | SPL ↑ |
|---|---|---|
| 0.0015 | 78.28 | 66.42 |
| 0.002 | 78.30 | 66.44 |
| **0.0025** | **78.32** | **66.47** |
| 0.003 | 78.29 | 66.45 |
| 0.0035 | 78.27 | 66.43 |
| 0.005 | 78.25 | 66.39 |

We evaluate the sensitivity of the three uncertainty–related hyperparameters on R2R `val unseen` split: $\delta$ and $\eta$, which regulate geometric uncertainty, and $\varepsilon$, which governs semantic uncertainty. The default settings used in our agent are $\delta$= 0.0025, $\eta$= 0.1, and $\varepsilon$= 0.0025. As shown in Table 10, the agent maintains similar performance when varying $\delta$ within 0.0015–0.005, $\eta$ within 0.05–0.2, and $\varepsilon$ within 0.0015–0.005, demonstrating that our uncertainty estimation is stable over a broad range of parameter values.

### D.2 STATISTICAL SIGNIFICANCE TESTS

Table 11: **Statistical Significance Tests** on R2R `val unseen` split. We report the mean ± std, confidence intervals (CI), and paired t-test $p$-values over 5 runs.

| Agent | SR ↑ | | | SPL ↑ | | |
|---|---|---|---|---|---|---|
| | mean±std | CI | $p$-value | mean±std | CI | $p$-value |
| DUET [11] | $72.22 \pm 0.21$ | [72.08, 72.36] | $3.42 \times 10^{-12}$ | $60.41 \pm 0.27$ | [60.27, 60.56] | $6.60 \times 10^{-15}$ |
| BEVBert [15] | $75.82 \pm 0.25$ | [75.70, 75.95] | $8.91 \times 10^{-11}$ | $64.14 \pm 0.22$ | [64.01, 64.28] | $7.30 \times 10^{-10}$ |
| VER [17] | $76.37 \pm 0.18$ | [76.27, 76.47] | $1.58 \times 10^{-10}$ | $65.07 \pm 0.21$ | [64.91, 65.23] | $2.90 \times 10^{-13}$ |
| **Ours** | $\mathbf{78.32 \pm 0.13}$ | **[78.25, 78.39]** | – | $\mathbf{66.47 \pm 0.17}$ | **[66.39, 66.54]** | – |

To assess whether the performance improvements are statistically meaningful beyond random variation, we conduct significance tests on R2R `val unseen` split over 5 runs. For each agent, we report the mean and standard deviation, the confidence interval (CI), and paired t-test $p$-values against ours. As shown in Table 11, the improvements of our agent over DUET [11], BEVBert [15], and VER [17] are statistically significant (all $p < 0.05$ for both SR and SPL), confirming that the gains are not attributable to stochastic variance.

# E UNCERTAINTY DISCUSSION AND FAILURE CASE

In this section, we provide an extended discussion of our uncertainty formulation, present additional empirical analyses, and summarize representative failure cases.

## E.1 DISCUSSION OF OUR UNCERTAINTY

Table 12: **Robustness** to observation noise on R2R `val unseen` split. We evaluate an *epistemic only* variant (geometric + semantic), an *aleatoric only* variant (appearance), and our agent under increasing levels of Gaussian noise in RGB observations.

| Noise Level | Epistemic Only SR ↑ | Aleatoric Only SR ↑ | Ours SR ↑ |
|:---:|:---:|:---:|:---:|
| 0% | 77.05 | 76.86 | **78.32** |
| 10% | 76.78 | 76.80 | **78.09** |
| 20% | 76.53 | 76.77 | **78.12** |
| 30% | 75.93 | 76.81 | **77.98** |

In deep learning, uncertainty is typically categorized into two types: *epistemic* and *aleatoric* [87–89]. Epistemic uncertainty arises from a lack of knowledge or limited evidence in the model, whereas aleatoric uncertainty denotes irreducible randomness or inherent variability in the data that cannot be reduced by collecting more samples. In embodied navigation [90], epistemic uncertainty is typically associated with insufficient or unreliable perceptual evidence (*e.g.*, missing views or out-of-distribution observations), often leading to ambiguous target grounding around visually similar landmarks [91]. Aleatoric uncertainty captures irreducible ambiguity caused by partial observability, occlusions, clutter, or sensor noise, which makes traversability inherently uncertain [91, 92].

Under this taxonomy, our design is as follows. *i*) We interpret geometric and semantic uncertainty as epistemic uncertainty. These two arise from missing or ambiguous perceptual evidence, such as sparsely observed regions or visually similar landmarks. Because they can, in principle, be reduced by acquiring more views, they align with the notion of epistemic uncertainty. *ii*) We interpret appearance uncertainty as aleatoric uncertainty. It reflects the sensitivity of rendered observations to small local perturbations. This variability is intrinsic to the rendering or measurement process and cannot be eliminated even if additional scene cues are available, which aligns with aleatoric uncertainty.

In addition, we examine whether these components behave consistently with the above interpretations. We compare three variants on R2R `val unseen` split: an *epistemic only* variant that uses geometric and semantic uncertainty, an *aleatoric only* variant that uses appearance uncertainty, and our agent. We gradually inject 10%, 20%, and 30% Gaussian noise into RGB observations while keeping all other settings fixed. As shown in Table 12, two trends align with the intended distinction. *i*) The *epistemic only* variant degrades as noise increases, reflecting its dependence on the sufficiency and reliability of perceptual evidence. *ii*) The *aleatoric only* variant remains stable across noise levels, consistent with uncertainty that models inherent observation variability. Moreover, our agent remains robust under all noise levels and achieves the best overall performance.

## E.2 ANALYSIS OF APPEARANCE UNCERTAINTY

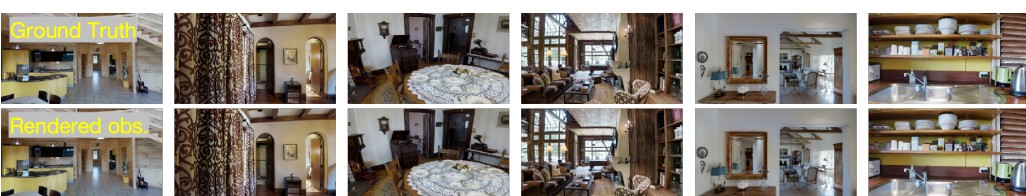

Figure 6: **Ground Truth *vs* Rendered Observations.** The renderings closely match the ground truth, supporting the Fisher-based appearance uncertainty proxy.

We provide visual comparisons between the rendered observations and the ground truth. Fig. 6 illustrates that the renderings closely match the ground truth, indicating that the residual term in the Hessian decomposition is negligible. Consequently, the Fisher Information serves as a reliable

proxy for appearance uncertainty and yields a faithful sensitivity signal, thereby corroborating the soundness of our design.

### E.3    EFFECTIVENESS OF UNCERTAINTY INFORMATION

Table 13: **Effectiveness of Uncertainty Information** on R2R `val unseen` split. The uncertainty information encoded in our 3D Value Map is rendered into a 2D panoramic uncertainty map and provided as an additional observation to support decision-making in existing agents.

| Agent | SR ↑ | SPL ↑ |
|---|---|---|
| DUET [11] | 72.22 | 60.41 |
| **DUET + 2D uncertainty map** | **73.52** | **61.43** |
| BEVBert [15] | 75.82 | 64.14 |
| **BEVBert + 2D uncertainty map** | **76.91** | **65.77** |
| VER [17] | 76.37 | 65.07 |
| **VER + 2D uncertainty map** | **77.45** | **65.94** |

To further verify that the estimated uncertainty information is beneficial for VLN, we apply our uncertainty cues to other agents. Since these agents adopt other forms of scene representation rather than 3DGS [93], we render the uncertainty encoded in our 3D Value Map into a 2D panoramic uncertainty map and provide it as an additional observation to DUET [11], BEVBert [15], and VER [17].

As shown in Table 13, this 2D uncertainty map consistently improves navigation performance across all three agents on R2R [1] `val unseen` split. For example, DUET improves from 72.22% to **73.52**% SR, while BEVBert and VER obtain similar gains of more than 1% in SR together with corresponding improvements in SPL. These results demonstrate that our uncertainty cues provide useful guidance for improving navigation decisions.

### E.4    FAILURE CASES

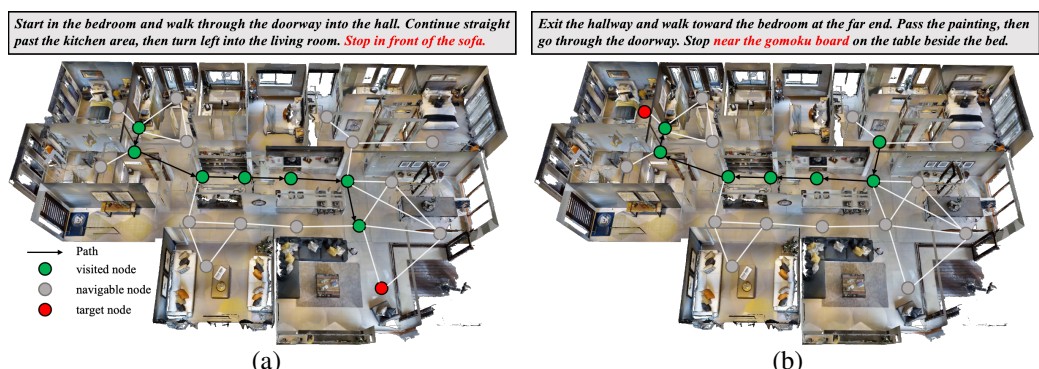

Figure 7: **Failure Cases.** (a) Our agent stops once *"the sofa"* comes into view, as the current observation already provides sufficient evidence of the target, creating confusion about whether further steps are required. (b) Our agent halts at the doorway instead of reaching *"the gomoku board"* near the bed, since the board lies inside the room and cannot be observed from the entrance, leaving the agent uncertain and leading to premature termination.

To illustrate the challenges our agent may still face, we present two representative failure cases. As shown in Fig. 7(a), although the instruction requires stopping at the sofa, the agent terminates as soon as *"the sofa"* enters its observation. This because the current view already provides sufficient evidence of the target, leaving the agent confused about whether perceiving *"the sofa"* is equivalent to reaching the intended stopping point. In addition, in Fig. 7(b), although the instruction requires *"stopping near the gomoku board by the bed"*, the agent halts at the doorway without entering the room. This because *"the board"* is not visible from its current viewpoint, leaving the agent uncertain about whether further exploration is necessary.

Table 14: **Quantitative Analysis** of uncertainty scaling on R2R `val unseen` split. We scale the estimated uncertainty values by factors in $\{1.0, 0.8, 0.6, 0.4, 0.2, 0\}$ and evaluate the agent in simple and complex scenes to analyze the influence of uncertainty on navigation performance.

| Uncertainty Scale | Simple Scenes SR ↑ | Complex Scenes SR ↑ |
|---|---|---|
| **1.0 (ours)** | **10/10** | **10/10** |
| 0.8 | 10/10 | 9/10 |
| 0.6 | 10/10 | 7/10 |
| 0.4 | 10/10 | 7/10 |
| 0.2 | 10/10 | 5/10 |
| 0 (No uncertainty) | 10/10 | 2/10 |

In addition, we further provide a quantitative analysis to examine how uncertainty affects navigation performance under different levels of perceptual ambiguity. Specifically, we select 20 representative scenes from the R2R `val unseen` split: 10 simple scenes (*e.g.*, open spaces, few obstacles, clear landmarks) and 10 complex scenes (*e.g.*, narrow spaces, occlusions, visually similar structures). For each episode, we run the agent while scaling the three uncertainty values by a factor in $\{1.0, 0.8, 0.6, 0.4, 0.2, 0\}$.

As shown in Table 14, performance in simple scenes remains consistently high across all scaling factors, indicating that these environments contain minimal perceptual ambiguity and rely little on uncertainty cues. In contrast, navigation performance in complex scenes drops progressively as the uncertainty values are suppressed. Removing the uncertainty information entirely reduces the success rate from **10/10** to 2/10, demonstrating that uncertainty cues play a critical role in guiding reliable navigation under challenging visual conditions.

## F    DISCUSSION

### F.1    LIMITATION AND FUTURE WORK

This work has several limitations that also highlight directions for future exploration. *i) Simulator Constraints.* Our framework is trained and evaluated in the static Matterport3D simulator [80], which omits real-world challenges such as moving objects, sensor noise, or actuation errors. Extending to dynamic and noisy environments will be crucial for deployment. *ii) Task Scope.* We focus on indoor VLN tasks (*i.e.*, R2R [1], RxR [27], REVERIE [28]). Applications to broader navigation domains, such as aerial VLN [94] or outdoor scenarios [95], remain unexplored. *iii) Environmental Coverage.* Our approach is primarily validated in structured indoor layouts. Future studies should examine its robustness in more cluttered, unstructured, or cross-domain environments. *iv) Predictive or Active Perception.* Our framework currently estimates perceptual uncertainty based solely on the available viewpoint, without actively acquiring additional evidence. Incorporating predictive view synthesis, such as world-model based future observation forecasting [24], or integrating active perception mechanisms [96] may allow the agent to select more informative viewpoints and thereby mitigate perceptual ambiguity. Exploring such predictive and action-guided perception strategies represents a promising direction for future research.

### F.2    TOWARD REAL-WORLD DEPLOYMENT

Although our experiments are conducted in simulation, transferring the proposed framework to real robots is an important direction. We outline several practical considerations and discuss how our design can be extended to address them.

**Sensor noise and uncertainty degradation.** To assess robustness to imperfect sensing, we inject 10–30% Gaussian noise into RGB observations and re-evaluate the agent on R2R `val unseen` split in Table. 12. The performance remains stable under these perturbations, suggesting that the uncertainty estimates remain stable under moderate sensor noise.

**Dynamic objects and time-varying geometry.** Real environments often contain moving objects and non-static geometry. Our Semantic Gaussian Map can be coupled with dynamic 3D Gaussian Splatting pipelines [97, 98], which continuously update Gaussians as the scene changes. In such a setup, both the scene representation and its associated uncertainties are updated online, enabling the agent to react to newly introduced ambiguity.

**Actuation errors and control dynamics.** Actuation errors affect the executed robot pose rather than the uncertainty estimation itself. For real-world deployment, our 3D Value Map can be integrated with standard closed-loop control and localization modules (*e.g.*, visual odometry [99] or SLAM [93]), so that pose uncertainty and perceptual uncertainty are jointly considered when planning reliable trajectories.

### F.3    SOCIAL IMPACT

This work introduces an uncertainty-aware framework for Vision–Language Navigation. By explicitly modeling geometric, semantic, and appearance uncertainties, the agent learns to interpret environments not only in terms of structure and semantics but also in terms of reliability. This design strengthens decision-making and improves navigation performance across multiple benchmarks. Beyond quantitative gains, the framework highlights the importance of uncertainty modeling for embodied AI, suggesting that safer, more interpretable, and reliability-aware navigation systems can be developed for broader real-world applications. We hope that this perspective will inspire future research on integrating uncertainty into embodied reasoning and planning.

### F.4    TERMS OF USE, PRIVACY, AND LICENSE

Matterport3D [80], R2R [1], RxR [27], and REVERIE [28] are available for non-commercial research purposes.

**Use of Large Language Models.** We did not use any large language models in this work.

