# OpenReview forum: "Uncertainty-Aware Gaussian Map for Vision-Language Navigation"
_ICLR.cc/2026/Conference — ICLR 2026 Poster_

### Official Review · Reviewer_ZYEA · 2025-10-25

**Soundness:** 3
**Presentation:** 2
**Contribution:** 3
**Rating:** 6
**Confidence:** 3

**Summary:**

This paper presents an uncertainty-aware Vision-Language Navigation (VLN) framework that explicitly models and leverages perceptual uncertainty to improve the navigation decision making. The key contribution is constructing a Semantic Gaussian Map (SGM) using differentiable 3D Gaussian primitives from panoramic observations, then estimating three forms of uncertainty: geometric, semantic, and appearance uncertainty, which are then integrated into a unified 3D Value Map that encodes affordances and constraints, guiding more reliable navigation decisions. Empirically, the proposed method improves the performance across benchmarks including R2R, RxR, and REVERIE, demonstrating the method's effectiveness.

**Strengths:**

1. The motivation and problem identification is clear. The paper motivates why uncertainty matters in VLN through concrete examples (e.g., confusing similar doors, occlusion ambiguity) that existing agents struggle to handle.
2. The proposed framework is technically sound. The combination of 3D Gaussian Splatting with variational inference for uncertainty estimation is well-grounded. The mathematical formulation is rigorous and clearly presented. The uncertainty estimation addresses three types of uncertainty, providing a complete picture.
3. Experiments are well-conducted, validating the effectiveness of the proposed method. Comprehensive ablation study demonstrates each component's contribution. Runtime analysis shows acceptable overhead, eliminating the concern about  excessive computational resource consumption.

**Weaknesses:**

1. Limited hyperparameter sensitivity analysis. Only the pruning threshold $\tau_e$ and $\tau_{\alpha}$ are analyzed (in Table 5). Sensitivity of uncertainty-related hyperparameters like $\delta$ and $\eta$ for geometric uncertainty and $\epsilon$ for semantic uncertainty.
2. Theoretical grounding of uncertainty identification is weak. While Appendix E.1 briefly discusses epistemic vs. aleatoric uncertainty, the mapping of geometric/semantic to epistemic and appearance to aleatoric lacks rigorous justification. This categorization seems somewhat arbitrary.
3. While standard deviations are reported, no significance tests confirm whether improvements are statistically meaningful beyond variance.
4. It would be more persuasive if the authors could upload video demos.

**Questions:**

1. Could the authors provide a comprehensive hyperparameter sensitivity analysis?
2. Could the authors provide a stronger justification regarding the uncertainty categorization (mapping geometric, semantic, appearance to epistemic and aleatoric)?
3. Could the authors conduct significance tests to prove their improvements are statistically meaningful?
4. As mentioned in the Limitation and Future Work, since the major contributions of the work are made in simulation environments. Any thoughts on addressing expected transfer problems to the real world? Like potential uncertainty estimate degradation with real sensor noise, dynamic objects, or actuation errors.

---

> ### Author Response · Authors · 2025-11-26
> **Responses to Reviewer ZYEA (1/4)**
>
> We sincerely thank Reviewer ZYEA for the valuable suggestions. Point-by-point responses are provided below, and all associated revisions have been incorporated into the revised manuscript and highlighted in **red**. All references (*e.g.*, Lines, Tables, Sections) correspond to the updated version.
>
> ---
>
> **Q1:** *"Limited hyperparameter sensitivity analysis..."*
>
> **A1:** Good suggestion! **First**, we would like to clarify that the default values of the three hyperparameters are $δ$ = 0.0025, $η$ = 0.1, and $ε$ = 0.0025. **Second**, we conduct sensitivity analysis for $δ$, $η$, and $ε$ on R2R val unseen split. For $δ$, the results below show that varying it between 0.0015 and 0.005 produces only marginal changes.
> | $δ$       | R2R *val unseen* SR ↑    | R2R *val unseen* SPL ↑  |
> |:-----------------:|:-------:|:-------:|
> | 0.0015               | 78.27     | 66.41     |
> | 0.002                | 78.29     | 66.44     |
> | **0.0025 (default)** | **78.32** | 66.47 |
> | 0.003                | 78.30     | 66.46     |
> | 0.0035               | 78.28     | **66.48**     |
> | 0.005                | 78.25     | 66.40     |
>
> For $η$, we provide detailed results as follows, and find that it has minimal impact on performance when varied within the range of 0.05-0.2.
> | $η$         | R2R *val unseen* SR ↑        | R2R *val unseen* SPL ↑      |
> |:-------------------:|:-----------:|:-----------|
> | 0.05              | 78.30     | 66.45     |
> | **0.1 (default)** | **78.32** | **66.47** |
> | 0.15              | 78.30     | 66.44     |
> | 0.2               | 78.28     | 66.41     |
>
> For $ε$, more experimental results presented below demonstrate that performance remains stable when its value falls between 0.0015 and 0.005.
> | $ε$       | R2R *val unseen* SR ↑   | R2R *val unseen* SPL ↑   |
> |:-----------------:|:-------:|:-------:|
> | 0.0015               | 78.28     | 66.42     |
> | 0.002                | 78.30     | 66.44     |
> | **0.0025 (default)** | **78.32** | **66.47** |
> | 0.003                | 78.29     | 66.45     |
> | 0.0035               | 78.27     | 66.43     |
> | 0.005                | 78.25     | 66.39     |
>
> In the revised version, we include these hyperparameter sensitivity experiments in Appendix E.1 (Lines 1031-1048, Table 10).

---

> ### Author Response · Authors · 2025-11-26
> **Responses to Reviewer ZYEA (2/4)**
>
> **Q2:** *"Theoretical grounding of uncertainty identification is weak..."*
>
> **A2:** Thank you for the insightful comment. We clarify our uncertainty categorization from both a conceptual and an empirical perspective.
>
> **From a conceptual perspective.** In deep learning [1], epistemic uncertainty arises from a lack of knowledge or limited evidence in the model, and aleatoric uncertainty denotes irreducible randomness or inherent variability in the data that cannot be reduced by collecting more samples. In embodied navigation [2], epistemic uncertainty is typically associated with insufficient or unreliable perceptual evidence (*e.g.*, missing views or out-of-distribution observations), while aleatoric uncertainty captures irreducible ambiguity caused by partial observability.
>
> Under this taxonomy, our design is as follows: i) Geometric and semantic uncertainty stem from missing or ambiguous perceptual evidence, such as visually similar landmarks. These uncertainties can, in principle, be reduced by acquiring more informative viewpoints, and we therefore interpret them as epistemic uncertainty. ii) Appearance uncertainty measures the sensitivity of rendered observations to small local perturbations. This variability is intrinsic to the rendering or measurement process and cannot be eliminated even with additional scene cues. Hence, we interpret it as aleatoric uncertainty.
>
> **From an empirical perspective.** We also examine whether the two components behave consistently with these interpretations. We compare three variants on R2R val unseen split: an epistemic-only variant (using geometric and semantic uncertainty), an aleatoric-only variant (using appearance uncertainty), and our agent. We gradually inject 10%, 20%, and 30% Gaussian noise into the RGB observations while keeping all other settings fixed. As shown in the table below, two trends align with the intended distinction. i) As RGB noise increases, the epistemic-only variant shows a gradual degradation, which aligns with epistemic uncertainty being more dependent on the quality of perceptual evidence. ii) The aleatoric-only variant remains relatively stable across noise levels, indicating that this component is less affected by additional observation noise, as expected for aleatoric uncertainty. In addition, our agent, which integrates both components, remains robust across all noise levels and achieves the best overall performance.
> | Noise Level  |Epistemic-Only Variant (R2R *val unseen* SR↑) |Aleatoric-Only Variant (R2R *val unseen* SR↑)  | our agent (R2R *val unseen* SR↑) |
> |:--------------:|:--------------------------------:|:----------------------------:|:-----------:|
> | 0%          |               77.05           |             76.86         |   **78.32** |
> | 10%          |                76.78           |             76.80          |     **78.09** |
> | 20%          |                76.53           |             76.77          |     **78.12** |
> | 30%          |                75.93           |             76.81          |     **77.98** |
>
> In the revised version, we expand Appendix F.1 to strengthen both the theoretical and empirical grounding of our uncertainty formulation (Lines 1086-1118, Table 12).
>
> [1] CertainlyUncertain: A Benchmark and Metric for Multimodal Epistemic and Aleatoric Awareness. In ICLR 2025.
>
> [2] EVORA: Deep Evidential Traversability Learning for Risk-Aware Off-Road Autonomy. IEEE Transactions on Robotics 2024.

---

> ### Author Response · Authors · 2025-11-26
> **Responses to Reviewer ZYEA (3/4)**
>
> **Q3:** *"...no significance tests confirm whether improvements are statistically meaningful beyond variance."*
>
> **A3:** Thank you for the valuable suggestion. We conduct statistical significance tests on R2R val unseen split over 5 runs, and report the mean ± std, confidence intervals (CI), and p-values computed using paired t-tests. As shown in the table below, the improvements of our agent over others [1-3] are statistically significant (p < 0.05).
> | Agent | mean ± std (R2R *val unseen* SR ↑) | CI (R2R *val unseen* SR ↑) | p-value (R2R *val unseen* SR ↑)| mean ± std (R2R *val unseen* SPL ↑) | CI (R2R *val unseen* SPL ↑)| p-value (R2R *val unseen* SPL ↑)|
> |:--------:|:-----------------:|:--------:|:---------:|:------------------:|:--------:|:---------:|
> | DUET [1]     | 72.22 ± 0.21 | [72.08, 72.36] | 3.42 × 10$^{-12}$ | 60.41 ± 0.27 | [60.27, 60.56] | 6.60 × 10$^{-15}$ |
> | BEVBert [2]  | 75.82 ± 0.25 | [75.70, 75.95] | 8.91 × 10$^{-11}$  | 64.14 ± 0.22 | [64.01, 64.28] | 7.30 × 10$^{-10}$ |
> | VER [3]      | 76.37 ± 0.18 | [76.27, 76.47] | 1.58 × 10$^{-10}$ | 65.07 ± 0.21 | [64.91, 65.23] | 2.90 × 10$^{-13}$ |
> | **Ours**     | **78.32 ± 0.13** | **[78.25, 78.39]** | $-$ | **66.47 ± 0.17** | **[66.39, 66.54]** | $-$ |
>
> In the revised version, we add the statistical significance tests to Appendix E.2 (Lines 1050-1064, Table 11).
>
> [1] Think global, act local: Dual-scale graph transformer for vision-and-language navigation. In CVPR 2022.
>
> [2] BEVBert: Multimodal map pre-training for language-guided navigation. In ICCV 2023.
>
> [3] Volumetric environment representation for vision-language navigation. In CVPR 2024.
>
> ---
>
> **Q4:** *"...Any thoughts on addressing expected transfer problems to the real world?..."*
>
> **A4:** We fully agree that transferring to real-world settings is an important next step, and we appreciate the suggestions on potential failure modes. While our agent is currently evaluated in simulation, we have taken initial steps to probe robustness and outline how it can be extended to real robots. **i) Sensor noise and uncertainty degradation.** To examine robustness to imperfect sensing, we inject 10–30% Gaussian noise into the RGB observations and re-evaluate our agent on R2R val unseen split (Table in Q2). The performance of our agent remains stable under these perturbations, which indicates that it is, to some extent, resilient to moderate sensor noise. **ii) Dynamic objects and time-varying geometry.** For real-world deployment, we plan to couple the Semantic Gaussian Map with a real-time dynamic 3DGS pipeline [1, 2], which updates Gaussians to reflect moving objects and time-varying geometry. In this setup, both the Gaussians and their associated uncertainties are updated online, allowing the agent to track scene changes and respond to the resulting ambiguity. **iii) Actuation errors and control dynamics.** Actuation errors originate at the control layer and therefore do not directly affect the perceptual uncertainty estimates produced by our agent. They can, however, cause discrepancies between intended and executed poses, which influence how observations are incorporated into the map. In a real robot system, our 3D Value Map could be integrated with standard closed-loop control and localization modules (*e.g.*, visual odometry [3] or SLAM [4]) so that pose uncertainty and perceptual uncertainty are considered jointly when planning safe trajectories.
>
> In the revised version, we add an extended discussion on real-world deployment considerations in Appendix G.2 (Lines 1261-1277).
>
> [1] Deformable 3D Gaussians for High-Fidelity Monocular Dynamic Scene Reconstruction. In CVPR 2024.
>
> [2] 4D Gaussian Splatting for Real-Time Dynamic Scene Rendering. In CVPR 2024.
>
> [3] Reinforcement Learning Meets Visual Odometry. In ECCV 2024.
>
> [4] SplaTAM: Splat Track & Map 3D Gaussians for Dense RGB-D SLAM. In CVPR 2024.

---

> ### Author Response · Authors · 2025-11-26
> **Responses to Reviewer ZYEA (4/4)**
>
> **Q5:** *"It would be more persuasive if the authors could upload video demos."*
>
> **A5:** Thank you for the suggestion. Fig. 4 provides a step-wise visualization of the agent’s trajectory, showing the constructed Semantic Gaussian Map, the rendered observations, and the aggregated uncertainty map at each navigation step. These visualizations illustrate how the agent grounds the instruction throughout the trajectory and how uncertainty highlights visually ambiguous regions such as reflective surfaces, chandeliers, and repetitive structures (Lines 442-447). We agree that video demos are an effective way to illustrate the agent’s behavior, and we will definitely provide full video demonstrations on the project page in the final version.
>
> ---
>
> **Summary of Responses to Reviewer ZYEA**
>
> Your suggestions enabled us to refine and strengthen the paper in several key aspects:
> 1. Introduced the hyperparameter sensitivity experiments (Appendix E.1).
> 2. Expanded the theoretical and empirical grounding of our uncertainty formulation (Appendix F.1).
> 3. Incorporated statistical significance tests to support the reliability of the reported improvements (Appendix E.2).
> 4. Added an extended discussion on real-world deployment considerations (Appendix G.2).
>
> All corresponding revisions are highlighted in **red** throughout the revised version. Thank you again for your valuable feedback, and we welcome further discussions and insights.

---

> > ### Comment · Reviewer_ZYEA · 2025-11-27
> >
> > I appreciate the authors for their high-quality point-by-point responses. Their complementary experimental results on hyperparameter sensitivity analysis demonstrates the robustness of the proposed method against the uncertainty-related hyperparameter selection. The statistical significance tests underscore the superiority of the proposed method over the baselines. The conceptual clarification and empirical justification address my question regarding the uncertainty categorization. The future plan for transferring to real-world seems promising. Given the comprehensive responses from the authors and the revised version of the paper, I am happy to raise my score accordingly.

---

> > > ### Author Response · Authors · 2025-11-27
> > > **Thanks for your recognition and positive feedback**
> > >
> > > Thank you very much for your valuable time and thoughtful review. We are glad that our responses and the revised paper addressed your concerns, and we sincerely appreciate your updated score. Thank you again for helping us refine this work.

---

### Official Review · Reviewer_XCWv · 2025-10-28

**Soundness:** 3
**Presentation:** 3
**Contribution:** 3
**Rating:** 6
**Confidence:** 4

**Summary:**

The paper proposes an Uncertainty-Aware Gaussian Map (UAGM) framework for Vision-Language Navigation (VLN). The method represents the environment as a 3D Gaussian field and explicitly models three types of perceptual uncertainty—geometric, semantic, and appearance—via variational inference and Fisher-information-based estimation. These uncertainty estimates are aggregated into a 3D Value Map used by the navigation policy. The approach aims to improve robustness and interpretability by allowing the agent to reason about uncertain visual observations. Experiments on R2R, RxR, and REVERIE show consistent performance gains over prior VLN baselines, and qualitative visualizations demonstrate uncertainty-aware perception.

**Strengths:**

1. The paper introduces an explicit Gaussian-based 3D uncertainty representation, offering a principled way to quantify perception uncertainty in VLN.
2. The theoretical formulation (variational inference + Fisher information) is well motivated and mathematically sound.
3. Experiments are comprehensive, spanning three benchmarks with ablations and qualitative visualizations.
4. The paper is generally clear and technically solid, with detailed appendices.

**Weaknesses:**

1. The claim of being the first to introduce uncertainty modeling in VLN is somewhat overstated. Prior work such as LLM as Copilot for Coarse-grained Vision-Language Navigation (ECCV 2024)[1] already explores confidence or uncertainty-related reasoning. The main novelty here lies in adopting a Gaussian-based uncertainty representation, rather than introducing the uncertainty concept itself. Moreover, the estimated uncertainty is not explicitly incorporated into decision-making—it serves more as an auxiliary feature than a truly uncertainty-guided policy.
2. The paper employs ambitious terminology (e.g., 3D Value Map, Unified Uncertainty Grounding), which may slightly overstate conceptual novelty.
3. Although the appendix includes qualitative failure cases, the analysis remains superficial. There is no quantitative study demonstrating how uncertainty correlates with navigation errors or ambiguous observations.
4. The latest compared method is VER (CVPR 2024), which is relatively dated. It would strengthen the paper to include comparisons with more recent works such as [2][3][4].

References

[1]  LLM as Copilot for Coarse-grained Vision-Language Navigation, ECCV 2024.
[2]  Do Visual Imaginations Improve Vision-and-Language Navigation Agents?, CVPR 2025.
[3]  COSMO: Combination of Selective Memorization for Low-cost Vision-Language Navigation, ICCV 2025.
[4]  Bootstrapping Language-Guided Navigation Learning with Self-Refining Data Flywheel, ICLR 2025.

**Questions:**

Have the authors analyzed whether high predicted uncertainty can be used to predict or prevent navigation failure?

---

> ### Author Response · Authors · 2025-11-26
> **Responses to Reviewer XCWv (1/3)**
>
> We sincerely thank Reviewer XCWv for the insightful comments. We address each point in detail below, and all corresponding revisions have been incorporated into the updated manuscript and highlighted in **red**. All references (*e.g.*, Lines, Tables, Sections) refer to the revised version.
>
> ---
>
> **Q1-1:** *"The claim of being the first to introduce uncertainty modeling in VLN is somewhat overstated."*
>
> **A1-1:** We apologize for the confusion. We respectfully clarify that we do **not** claim to be "the first to introduce uncertainty modeling in VLN." Our intention lies in leveraging perceptual uncertainty as a spatial signal for navigation. Specifically, we estimate geometric, semantic, and appearance uncertainties directly from the agent’s observations and integrate them into a unified 3D Value Map that informs decision-making (Lines 051-052).
>
> In the revised version, we more clearly articulate the scope and role of our perceptual uncertainty and discuss other uncertainty-based VLN agents in the Related Work (Lines 104–107).
>
> ---
>
> **Q1-2:** *"Prior work such as LLM as Copilot for VLN-Copilot [1] already explores confidence or uncertainty-related reasoning."*
>
> **A1-2:** Thank you for this valuable suggestion. The uncertainty explored in our work is fundamentally distinct from that in VLN-Copilot [1]. VLN-Copilot [1] models **decision-level** uncertainty by computing a confusion score from the action probability distribution, and uses this score as a trigger to invoke external LLM assistance. In contrast, our work focuses on **perception-level** uncertainty, which captures geometric, semantic, and appearance unreliability arising directly from the observation rather than in the policy output (Lines 196-200).
>
> In the revised version, we add VLN-Copilot [1] to the Related Work (Sec. 2) and explicitly distinguish its decision-level uncertainty modeling from our perception-level formulation (Lines 104–107).
>
> [1] LLM as Copilot for Coarse-grained Vision-Language Navigation. In ECCV 2024.
>
> ---
>
> **Q1-3:** *"...the estimated uncertainty ... serves more as an auxiliary feature than a truly uncertainty-guided policy."*
>
> **A1-3:** Sorry for the confusion. We also clarify that the estimated uncertainties are **not** an auxiliary feature. They are embedded in the scene representation together with each Gaussian’s geometric and semantic attributes (Eq. 10). After projection into the feature space, the representation jointly encodes the scene structure and its perceptual confidence. The policy network operates on this representation, and its decisions are therefore directly conditioned on the uncertainty information as well (Eq. 11). In addition, removing the uncertainty information while keeping all other components unchanged consistently degrades navigation performance (Table 6), indicating that these uncertainty cues play an essential role in the decision-making process.
>
> In the revised version, we add details explaining how uncertainty influences the policy decisions in Sec. 3.3 (Lines 275-280).

---

> ### Author Response · Authors · 2025-11-26
> **Responses to Reviewer XCWv (2/3)**
>
> **Q2:** *"...ambitious terminology (e.g., 3D Value Map, Unified Uncertainty Grounding), which may slightly overstate conceptual novelty."*
>
> **A2:** Sorry for the confusion. **First**, the term value map is not specific to our work and is broadly used across 3D scene reasoning and robotics, such as affordance fields [1], cost maps [2], and traversability [3] or reliability maps [4]. These maps generally refer to a spatial field where each element encodes task-relevant signals that guide downstream decision-making. In prior work, for example, VoxPoser [5] defines a voxel-based value map in which each voxel stores affordance or constraint values (*e.g.*, graspability or reachability) generated through LLM-based reasoning for manipulation. Inspired by these formulations, our 3D Value Map realizes a value field on top of the Semantic Gaussian Map, where each Gaussian is augmented with geometric, semantic, and appearance uncertainty estimates (Eq. 10). Together, these uncertainties provide unified reliability cues that can be naturally interpreted as affordances and constraints for navigation. **Second**, the term Unified Uncertainty Grounding refers to the **process** of embedding three forms of perceptual uncertainty into a single spatial field. This step ensures that geometric, semantic, and appearance reliability cues are encoded within a consistent spatial structure rather than treated as isolated attributes. The resulting 3D Value Map thus serves as a unified perceptual-reliability field, where uncertainty functions as an affordance- or constraint-like signal that complements the geometric and semantic features used for navigation.
>
> In the revised version, we add a more detailed explanation of the 3D Value Map in Sec. 3.3 (Lines 260–266).
>
> [1] An interactive navigation method with effect-oriented affordance. In CVPR 2024.
>
> [2] Learning barrier functions with memory for robust safe navigation. IEEE Robotics and Automation Letters 2021.
>
> [3] EVORA: Deep Evidential Traversability Learning for Risk-Aware Off-Road Autonomy. IEEE Transactions on Robotics 2024.
>
> [4] Open-fusion: Real-time open-vocabulary 3d mapping and queryable scene representation. In ICRA 2024.
>
> [5] VoxPoser: Composable 3D Value Maps for Robotic Manipulation with Language Models. In CoRL 2023.
>
> ---
>
> **Q3:** *"...There is no quantitative study demonstrating how uncertainty correlates with navigation errors or ambiguous observations."*
>
> **A3:** Good suggestion! We select 20 representative scenes from R2R val unseen split: 10 simple scenes (*e.g.*, open spaces, few obstacles, clear landmarks) and 10 complex scenes (*e.g.*, narrow spaces, occlusions, visually similar structures). For each episode, we run our agent while scaling three uncertainty values by a factor in {1.0, 0.8, 0.6, 0.4, 0.2, 0}. The results are shown in the table below. In the simple scenes, perceptual ambiguity is minimal, and the performance remains consistently high (**10/10**) across all scaling factors. In the complex scenes, performance degrades progressively as the uncertainty information is suppressed, indicating that uncertainty cues are crucial for reliable navigation in perceptually challenging environments.
> | Uncertainty Scale|Simple Scenes (R2R *val unseen* SR ↑)| Complex Scenes (R2R *val unseen* SR ↑)|
> | :--------------------: | :----------------: | :-----------------: |
> | **1.0 (ours)**            | **10/10**              | **10/10**                |
> | 0.8                   | 10/10               | 9/10                 |
> | 0.6                   | 10/10               | 7/10                 |
> | 0.4                   | 10/10               | 7/10                 |
> | 0.2                   | 10/10               | 5/10                 |
> | 0 (No Uncertainty)    | 10/10               | 2/10                 |
>
> In the revised version, we include this quantitative analysis of how uncertainty scaling affects navigation performance in Appendix F.4 (Lines 1188–1209, Table 14).

---

> ### Author Response · Authors · 2025-11-26
> **Responses to Reviewer XCWv (3/3)**
>
> **Q4:** "*...include comparisons with more recent works such as DUET-Imagine [2], COSMO [3], and SRDF [4].*"
>
> **A4:** Thank you for the suggestion. We include the comparisons with DUET-Imagine [2], COSMO [3], and SRDF [4] in the table below. On R2R val unseen split, our agent outperforms DUET-Imagine [2] by **6**% SR and **6**% SPL, and surpasses COSMO [3] by **5**% SR and **5**% SPL. These margins indicate that our agent navigates more reliably on standard VLN instructions. Moreover, on REVERIE val unseen split, our agent improves RGS by **4.68**% and RGSPL by **3.76**% over DUET-Imagine [2]. These improvements demonstrate that our agent exhibits stronger capability in accurate navigation and reliable object grounding. In addition, although SRDF [4] achieves higher SR/SPL on R2R, these results are driven by **substantially larger additional supervision**, including over 20M synthetic instructions for 2.9M trajectories across 800+ environments.
> |  Agent   |R2R *val unseen* SR ↑ |R2R *val unseen* SPL ↑|REVERIE *val unseen* RGS ↑ |REVERIE *val unseen* RGSPL ↑|
> |:------:|:----------------:|:---------:|:---------:|:---------:|
> | DUET-Imagine [2]| 72    | 60    | 32.97    | 23.25    |
> | COSMO [3]       | 73    | 61    | $-$    | $-$    |
> | SRDF [4]        | 86    | 79    | $-$    | $-$    |
> |**Ours**         | **78**| **66**| **37.65**    | **27.01**    |
>
> In the revised version, we incorporate all three agents into the Related Work (Sec. 2), and include DUET-Imagine [2] and COSMO [3] in Tables 1 and 2 for comparison.
>
> [2] Do Visual Imaginations Improve Vision-and-Language Navigation Agents? In CVPR 2025.
>
> [3] COSMO: Combination of Selective Memorization for Low-cost Vision-Language Navigation. In ICCV 2025.
>
> [4] Bootstrapping Language-Guided Navigation Learning with Self-Refining Data Flywheel. In ICLR 2025.
>
> **Q5:** *"...whether high predicted uncertainty can be used to predict or prevent navigation failure?"*
>
> **A5:** Yes. **Qualitatively**, as shown in Fig. 1, areas with high uncertainty, such as ambiguous doorways or occluded corners, often coincide with locations where navigation errors tend to occur. In our agent, these regions appear as Gaussians with large uncertainty estimates in the 3D Value Map. This enables the agent to identify them as unreliable parts of the scene and to navigate more cautiously around them. **Quantitatively**, the analysis in Q3 also supports this point. When we progressively reduce the influence of uncertainty in perceptually complex scenes, the number of successful cases drops from 10 to 2, while performance in simple scenes remains unchanged. This demonstrates that uncertainty is beneficial in perceptually challenging scenes and contributes to more reliable navigation.
>
> ---
>
> **Summary of Responses to Reviewer XCWv**
>
> Your suggestions substantially strengthened the revised version in several aspects:
> 1. Clarified the distinction between our perception-level uncertainty formulation and prior decision-level approaches (Sec. 2).
> 2. Detailed how uncertainty information influences policy decisions (Sec. 3.3).
> 3. Expanded the explanation of the 3D Value Map (Sec. 3.3).
> 4. Added an analysis of how uncertainty scaling affects navigation performance (Appendix F.4).
> 5. Integrated the discussion of more recent VLN agents (Sec. 2).
>
> All corresponding revisions are highlighted in **red** throughout the revised version. Thank you again for your insightful comments, and we welcome further discussions and insights.

---

> ### Author Response · Authors · 2025-11-28
> **Looking forward to your further discussion**
>
> Dear Reviewer XCWv,
>
> Thank you again for your kind review and comments. We hope our responses and the revised paper have addressed your concerns. If any further clarification would be helpful, we would be glad to provide it. We sincerely hope that we will be able to use the remaining time to engage in further dialogue with domain experts to enhance the quality of our work. We truly appreciate your time and feedback.
>
> Thanks again, Authors

---

### Official Review · Reviewer_BefP · 2025-10-30

**Soundness:** 2
**Presentation:** 3
**Contribution:** 3
**Rating:** 4
**Confidence:** 4

**Summary:**

To address the perceptual uncertainty in robot navigation, this paper formalizes three forms of uncertainty, including geometric, semantic, and appearance uncertainty, enabling more informed decision-making. Specifically, a Semantic Gaussian Map (SGM) based on 3D Gaussian primitives is constructed to encode both the geometric structure and semantic content of environments. Then, three uncertainties are estimated: 1) geometric uncertainty is modeled through variational perturbations of Gaussian position and scale to assess structural reliability; 2) semantic uncertainty is estimated by perturbing the semantic attributes of Gaussians to reveal ambiguous interpretations; 3) appearance uncertainty is quantified by Fisher Information to measure the sensitivity of rendered observations to Gaussian-level variations. Then, a unified 3d Value Map is composed to ground these uncertainties as affordances and constraints, thereby guiding informed and more reliable trajectory planning. Extensive experiments on VLN benchmarks demonstrate the effectiveness of the proposed method.

**Strengths:**

1)	This paper focuses on a valuable but usually ignored research topic, i.e., the perceptual uncertainty in robot navigation, and systematically formalizes three forms of perceptual uncertainties, including geometric, semantic, and appearance uncertainty, which reveals a new improvement direction for robot navigation.
2)	Extensive experiments are constructed to demonstrate the effectiveness of the proposed uncertainties.
3)	This paper is well-written and easy to follow.

**Weaknesses:**

1）	Could the recognized perceptual uncertainty help to further lower the uncertainty? How do you improve the confidence/reliability? Will it help with robot navigation?
2）	In the action prediction of the 3D Value Map, all the representations of each Gaussian are projected into a feature vector, and then the aggregated representation is utilized to predict candidate nodes. So the uncertainties indirectly and aggregately impact action prediction, does it need to compute the uncertainty of each Gaussian?
3）	According to the results in Tables 1 and 2, the improvement of the proposed method is marginal over the state-of-the-art, especially on the SR, SPL.
4）	According to the results in Table 6, it seems the three uncertainties do improve the baseline method, but the baseline is obviously outperformed by the SOTA methods. Could the proposed method still work on a better baseline?

**Questions:**

Please try to address the weaknesses.

---

> ### Author Response · Authors · 2025-11-26
> **Responses to Reviewer BefP (1/2)**
>
> We sincerely thank Reviewer BefP for the constructive feedback. We provide point-to-point responses below, and all corresponding revisions have been incorporated into the revised paper and highlighted in **red**. All references (*e.g.*, Lines, Tables, Sections) correspond to the updated version.
>
> ---
>
> **Q1:** *"Could the recognized perceptual uncertainty help to further lower the uncertainty? How do you improve the confidence/reliability? Will it help with robot navigation?"*
>
> **A1:** Thank you for the thoughtful question. **First**, our agent is designed to estimate and use the perceptual uncertainty that reflects the inherent ambiguity in its observations, rather than to artificially reduce it (Lines 051–052). At a fixed waypoint, these uncertainties indicate which parts of the scene are ambiguous given the current view (*e.g.*, occlusion or visually similar structures), and such ambiguity cannot be resolved without obtaining additional evidence (Lines 046–049, Fig. 1). **Second**, explicitly recognizing uncertainty is already useful in two ways. i) In the 3D Value Map, each Gaussian encodes both its geometric and semantic attributes and the estimated perceptual uncertainty (Lines 270-271). This makes the perceptual space explicitly encode uncertainty-aware affordance cues, providing more informed guidance for action selection (Lines 272-273). ii) As the agent moves, it receives new panoramic observations and constructs a new 3D Value Map at each waypoint. Regions that are ambiguous at the previous node may become more certain when viewed from a more informative location, while consistently uncertain areas remain marked as unreliable. This allows the agent to maintain a more complete global understanding of the environment and make more reliable decisions. **Third**, further reducing uncertainty would require additional information beyond the current observation, such as predicting future views with a world model [1] or actively selecting actions [2] to acquire more informative evidence. These methods can, in principle, resolve inherent ambiguity in observations, but they lie outside the scope of this work. We regard exploring such predictive or active-perception strategies as promising future work.
>
> In the revised version, we clarify the nature of perceptual uncertainty in Sec. 1 (Lines 071-072), refine Sec. 3.5 to better explain its global impact (Lines 313-315), and further expand the future work in Appendix G.1 (Lines 1252-1257).
>
> [1] Pathdreamer: A World Model for Indoor Navigation. In ICCV 2021.
>
> [2] Move to Understand a 3D Scene: Bridging Visual Grounding and Exploration for Efficient and Versatile Embodied Navigation. In ICCV 2025.
>
> ---
>
> **Q2:** *"...does it need to compute the uncertainty of each Gaussian?"*
>
> **A2:** Yes, computing uncertainty for each Gaussian is essential. **First**, uncertainty in our framework is a per-Gaussian attribute on the same level as geometry and semantics (Eq. 10). It forms a fundamental part of our scene representation, constituting a "Gaussian-based uncertainty representation" (Reviewer XCWv, Lines 270-271). This design allows the agent to establish a direct correspondence between local geometry and its associated uncertainty in its perceptual space. Consequently, the aggregated representation inherently preserves this fine-grained coupling, enabling the agent to make decisions directly informed by such structure-aware uncertainty (Lines 289-290, Eq. 11). **Second**, per-Gaussian uncertainty is required to preserve the fine-grained reliability information of the scene. After each Gaussian is projected into the feature space and the features are aggregated, different local uncertainty patterns can collapse into similar aggregated representations. For example, a region that is mostly reliable but contains a few ambiguous Gaussians can produce an aggregated feature similar to a region that is uncertain overall. Without per-Gaussian uncertainty, this difference disappears in the aggregated feature, and the final representation no longer reflects which parts of the scene are reliable. **Third**, per-Gaussian uncertainty does not introduce prohibitive overhead. By controlling the number of Gaussians through selective pruning (Lines 187–192), our agent achieves an effective trade-off between efficiency and performance (Table 5).
>
> In the revised version, we refine Sec. 3.3 to offer a more detailed explanation of our Gaussian representation (Lines 275-280).

---

> ### Author Response · Authors · 2025-11-26
> **Responses to Reviewer BefP (2/2)**
>
> **Q3:** *"According to the results in Tables 1 and 2, the improvement of the proposed method is marginal..."*
>
> **A3:** Our work focuses on improving the agent’s perception and understanding of its environment, and therefore we compare primarily with map-based VLN agents. For **Table** 1, REVERIE focuses on remote object grounding. The instructions specify the target object and its approximate location, and the agent must navigate to a suitable viewpoint and then ground the referred object precisely (Lines 357-359). On this challenging benchmark, our agent improves RGS and RGSPL by **3.94**% and **3.31**% on val unseen split, which are **substantial gains**. For example, VER (CVPR2024) is 1% lower than BEVBert (ICCV2023) (*i.e.*, its strongest comparison baseline) in RGS and 0.74% lower in RGSPL on this split (Table 1), yet such margins are still considered meaningful on REVERIE. For **Table** 2, our agent improves SR by **2**% and SPL by **1**% on R2R val unseen split. While numerically smaller than on REVERIE, these gains are still **non-trivial**. For example, VER (CVPR2024) is 1% higher than BEVBert (ICCV2023) in both SR and SPL on this split (Table 2), and such margins are typically regarded as significant on R2R. In addition, these points were also acknowledged by the other reviewers, such as "show consistent performance gains" (Reviewer XCWv) and "improves the performance across benchmarks" (Reviewer ZYEA).
>
> In the revised version, we update Sec. 4.2 to clarify the reported improvements (Lines 376-377, 408-412).
>
> [1] BEVBert: Multimodal map pre-training for language-guided navigation. In ICCV 2023.
>
> [2] Volumetric environment representation for vision-language navigation. In CVPR 2024.
>
> ---
>
> **Q4:** *"...Could the proposed method still work on a better baseline?"*
>
> **A4:** Yes, our method can work with other baselines. We rendered our per-Gaussian uncertainties into a 2D panoramic uncertainty map and integrated this map as an additional observation into DUET [1], BEVBert [2], and VER [3]. As shown in the table below, even this 2D uncertainty information consistently improves navigation performance, *e.g.*, from 72.22% to **73.52**% SR of DUET on R2R val unseen split.
> | |R2R *val unseen* SR ↑ |R2R *val unseen* SPL ↑ |
> |:------:|:------------:|:-:|
> | DUET [1]                         | 72.22 | 60.41 |
> | **DUET [1] + 2D uncertainty map**    | **73.52** | **61.43** |
> | BEVBert [2]                      | 75.82 | 64.14 |
> | **BEVBert [2] + 2D uncertainty map** | **76.91** | **65.77** |
> | VER [3]                          | 76.37 | 65.07 |
> | **VER [3] + 2D uncertainty map**     | **77.45** | **65.94** |
>
> In addition, we would like to clarify that the baseline in **Table 6** (Row #1) utilizes our Semantic Gaussian Map (SGM) for the 3D scene representation. As evidenced in **Table 4** (Row #1 vs. Row #2), employing SGM alone substantially improves the original DUET baseline (*e.g.*, from 72.22% to **76.21**% SR on R2R val unseen split).
>
> In the revised version, we clarify the baseline description in Table 6 (Lines 489-490), and add analysis on the effectiveness of uncertainty information in Appendix F.3 (Lines 1137-1158).
>
> [1] Think global, act local: Dual-scale graph transformer for vision-and-language navigation. In CVPR 2022.
>
> [2] BEVBert: Multimodal map pre-training for language-guided navigation. In ICCV 2023.
>
> [3] Volumetric environment representation for vision-language navigation. In CVPR 2024.
>
> ---
>
> **Summary of Responses to Reviewer BefP**
>
> Your feedback helped us substantially improve the paper in multiple aspects:
> 1. Clarified the nature and global evolution of perceptual uncertainty (Secs. 1, 3.5).
> 2. Expanded the discussion of future work (Appendix G.1).
> 3. Refined the explanation of our Gaussian representation (Sec. 3.3).
> 4. Specified the reported performance improvements (Sec. 4.2).
> 5. Detailed the baseline configuration of Table 6.
> 6. Added analysis on the effectiveness of uncertainty information (Appendix F.3).
>
> All corresponding revisions are highlighted in **red** throughout the revised version. Thank you again for your helpful comments, and we welcome further discussions and insights.

---

> ### Author Response · Authors · 2025-11-28
> **Looking forward to your further discussion**
>
> Dear Reviewer BefP,
>
> Thank you again for your kind review and comments. We hope our responses and the revised paper have addressed your concerns. If any further clarification would be helpful, we would be glad to provide it. We sincerely hope that we will be able to use the remaining time to engage in further dialogue with domain experts to enhance the quality of our work. We truly appreciate your time and feedback.
>
> Thanks again, Authors

---

### Author Response · Authors · 2025-11-26
**Summary of Our Responses & Paper Revisions**

We sincerely thank all reviewers for their valuable time and insightful feedback. We appreciate that the reviewers acknowledged the clarity of our motivation (Reviewers BefP, ZYEA), the soundness of our theoretical formulation (Reviewers XCWv, ZYEA), the well-conducted experiments (Reviewers BefP, XCWv, ZYEA), and the readability of the paper (Reviewers BefP, XCWv).

We have made our best effort to address the concerns and revise the paper accordingly. The major modifications are summarized as follows:

**Clarifications on Perceptual Uncertainty and the 3D Value Map**

1. Provided additional clarification on perceptual uncertainty and the Gaussian representation (Secs. 1, 3.3, 3.5).
2. Distinguished our perception-level uncertainty formulation from prior decision-level approaches (Sec. 2).
3. Explained more clearly how uncertainty information informs policy decisions (Sec. 3.3).
4. Refined the description of the 3D Value Map (Sec. 3.3).
5. Strengthened the theoretical and empirical grounding of our uncertainty taxonomy (Appendix F.1).

**Additional Experiments and Analyses**
1. Added hyperparameter sensitivity experiments (Appendix E.1).
2. Added statistical significance tests (Appendix E.2).
3. Added analysis of robustness to observation noise (Appendix F.1).
4. Added analysis on the effectiveness of uncertainty information (Appendix F.3).
5. Added analysis of uncertainty scaling (Appendix F.4).

**Expanded Discussions**
1. Integrated several recent VLN agents into the Related Work (Sec. 2).
2. Expanded the discussion of future work (Appendix G.1).
3. Added real-world deployment considerations (Appendix G.2)

The major revised contents in the revised version are highlighted in **red**. Point-for-point responses to specific comments are given in the following reviewer-specific responses. We welcome any further discussions and will address any remaining concerns.

---

### Author Response · Authors · 2025-12-02
**Summary of Reviews and Responses**

We sincerely appreciate the Area Chair for their effort in handling the unexpected issues during the review process, and we are grateful to all reviewers for their valuable time and insightful feedback. In response, we have provided detailed point-by-point clarifications and revised the manuscript accordingly. We summarize the key aspects of our work, the reviewers’ comments, and our responses as follows.

---

**Summary of This Work**

The paper proposes an Uncertainty-Aware Gaussian Map framework for VLN (Reviewer XCWv). The method represents the environment as a 3D Gaussian field and explicitly models three types of perceptual uncertainty—geometric, semantic, and appearance—via variational inference and Fisher-information–based estimation (Reviewer XCWv). A unified 3D Value Map is then composed to ground these uncertainties as affordances and constraints, thereby guiding more informed and reliable trajectory planning (Reviewer BefP). Empirically, the method improves performance across R2R, RxR, and REVERIE, demonstrating its effectiveness (Reviewer ZYEA).

---

**Summary of Reviewers’ Comments and Our Responses**

**Reviewer BefP (Rating 4 & Confidence 4)**

Reviewer BefP notes that the paper “focuses on a valuable but usually ignored research topic and reveals a new improvement direction for robot navigation.” The main concerns relate to: i) an alternative perspective on our uncertainty, focusing on minimizing it rather than utilizing it (q1); ii) the necessity of per-Gaussian uncertainty (q2); iii) the magnitude of performance improvement (q3); and iv) the generalizability of our uncertainty modeling to other baselines (q4).

In response, we i) clarify the role of perceptual uncertainty in our framework and expand the discussion of its potential extensions in future work (Secs. 1 & 3.5; Appendix G.1); ii) explain the necessity of per-Gaussian uncertainty and refine the description of our Gaussian-based representation (Sec. 3.3); iii) clarify the reported performance gains and strengthen the corresponding analysis (Sec. 4.2); and iv) add supplementary experiments to support the generalizability of our uncertainty modeling (Appendix F.3).

**Reviewer XCWv (Rating 6 & Confidence 4)**

Reviewer XCWv acknowledges the technical clarity and soundness of the paper, and highlights the comprehensiveness of the experiments and overall presentation. The main concerns relate to: i) the novelty of our uncertainty modeling and how the estimated uncertainty informs decision-making (q1); ii) the potentially overstated terminology used for the 3D Value Map (q2); iii) the need for a more quantitative study of failure cases (q3); and iv) the insufficient comparison against more recent VLN agents (q4).

In response, we i) clarify the distinction between our perception-level uncertainty formulation and prior decision-level approaches, and detail how uncertainty information influences policy decisions (Secs. 2 & 3.3); ii) expand the explanation of our 3D Value Map (Sec. 3.3); iii) add quantitative analysis on how uncertainty scaling relates to navigation performance (Appendix F.4); and iv) include more recent VLN agents in the Related Work and comparisons (Sec. 2; Tables 1 & 2).

**Reviewer ZYEA (Rating 6 & Confidence 3)**

Reviewer ZYEA recognizes the clarity of the motivation and problem formulation, the technical soundness of the proposed framework, and the well-conducted experimental evaluation.  The main concerns relate to: i) the limited hyperparameter sensitivity analysis (q1); ii) the insufficient theoretical grounding of the geometric, semantic, and appearance uncertainty formulation (q2); iii) the absence of statistical significance tests (q3); and iv) the need for deeper discussion regarding transfer to real-world scenarios (q4).

In response, we i) add a more comprehensive hyperparameter sensitivity analysis covering uncertainty-related parameters (Appendix E.1); ii) strengthen the theoretical justification of our geometric, semantic, and appearance uncertainty formulation (Appendix F.1); iii) include statistical significance tests (Appendix E.2); and iv) expand the discussion on real-world transfer challenges (Appendix G.2).

Following our response, Reviewer ZYEA states that the supplementary experiments and clarifications fully address their concerns and raises their score accordingly.

---

Detailed explanations are provided in the point-by-point responses, and all corresponding revisions are highlighted in **red** throughout the revised manuscript. Thank you again for your time and consideration.

---

### Meta-Review · Area_Chair_Uqm4 · 2026-01-14

**Summary:**

The paper proposes an Uncertainty-Aware Gaussian Map (UAGM) framework for Vision-Language Navigation (VLN). By constructing a 3D Gaussian field that incorporates three types of perceptual uncertainty, the method aims to enhance the reliability of navigation decision-making. During the initial review, reviewers generally acknowledged the soundness of the method's theoretical construction (variational inference and Fisher information) and the completeness of the experiments. However, the primary concerns centered on marginal performance improvement, Novelty and baselines, theoretical analysis. Based on the authors' exceptionally detailed rebuttal on additional experiments and method clarification, I believe the authors have convincingly addressed all major technical concerns. Reviewer ZYEA explicitly stated in the discussion phase that they would raise their score. Therefore, acceptance is recommended.

**Reviewer Concerns:**

The authors provided extensive supplementary experiments and analyses in their rebuttal, including statistical significance tests, additional baselines, clarification of novelty, and theoretical justification. These additions effectively address the above concerns.

Although the authors added a discussion on sensor noise and dynamic objects in the appendix, which satisfied Reviewer ZYEA, the deployment from simulation to real-world robots remains largely theoretical.

**Reviewer Scores:**

Reviewer ZYEA (Current: 6): Predicted: 6. The authors perfectly addressed their requirements regarding sensitivity analysis and significance testing.

Reviewer BefP (Current: 4): Predicted: 6. By demonstrating the general effectiveness of the method on other strong baselines, the authors directly resolved this reviewer's core concern about "marginal improvement."

Reviewer XCWv (Current: 6): Predicted: 6. The authors included the requested comparisons with the latest literature and provided quantitative ablation experiments (Uncertainty Scaling) to prove that the uncertainty module is not merely an auxiliary feature but is crucial for decision-making, thereby strengthening the paper’s quality.

---

### Decision · Program_Chairs · 2026-01-26

Accept (Poster)